# Aerosol above cloud direct radiative effect and properties in the Namibian region during AEROCLO-sA field campaign: 3MI airborne simulator and sun-photometer measurements.

Aurélien Chauvigné[1], Fabien Waquet[1], Frédérique Auriol[1], Luc Blarel[1], Cyril Delegove[1], Oleg Dubovik[1], Cyrille Flamant[2], Marco Gaetani[2,3,6], Philippe Goloub[1], Rodrigue Loisil[1], Marc Mallet[4], Jean-Marc Nicolas[1], Frédéric Parol[1], Fanny Peers[5], Benjamin Torres[1], and Paola Formenti[6].

[1] Univ. Lille, CNRS, UMR 8518 - LOA - Laboratoire d'Optique Atmosphérique, F-59000 Lille, France.
[2] LATMOS, UMR CNRS 8190, Sorbonne Université, Université Paris-Saclay, Institut Pierre Simon Laplace, Paris, France.
[3] Scuola Universitaria Superiore IUSS, Pavia, Italy.
[4] CNRM, Université de Toulouse, Météo-France, CNRS, Toulouse, France.
[5] CEMPS, University of Exeter, Exeter, EX4 4QE, UK.
[6] LISA, UMR CNRS 7583, Université Paris-Est-Créteil, Université de Paris, Institut Pierre Simon Laplace, Créteil, France.

*Correspondence to*: Aurélien Chauvigné (aurelien.chauvigne@univ-lille.fr)

**Abstract.** We analyse the airborne measurements of above-cloud aerosols from the Aerosols, radiation and clouds in Southern Africa (AEROCLO-sA) field campaign performed in Namibia during August and September 2017. The study aims to retrieve the aerosol above cloud Direct Radiative Effect (DRE) with well-defined uncertainties. To improve the retrieval of the aerosol and cloud properties, the airborne demonstrator of the Multi-viewing, Multi-channel, Multi-polarization (3MI) satellite instrument, called the Observing System Including PolaRisation in the Solar Infrared Spectrum (OSIRIS), was deployed on-board the Safire Falcon 20 aircraft during 10 flights performed over land, over the ocean and along the Namibian coast. The airborne instrument OSIRIS provides observations at high temporal and spatial resolutions for Aerosol Above Clouds (AAC) and cloud properties. OSIRIS was supplemented with the Photomètre Léger Aéroporté pour la surveillance des Masses d'Air version 2 (PLASMA2). The combined airborne measurements allow for the first time the validation of Aerosol Above Cloud (AAC) algorithms previously developed for satellite measurements. The variations of the aerosol properties are consistent with the different atmospheric circulation regimes observed during the deployment. Airborne observations typically show strong Aerosol Optical Depth (AOD, up to 1.2 at 550 nm) of fine mode particles from biomass burning (extinction Angström exponent varying between 1.6 and 2.2), transported above bright stratocumulus decks (mean cloud top around 1 km above mean sea level) with Cloud Optical Thickness (COT) up to 35 at 550 nm. The above-cloud visible AOD retrieved with OSIRIS agrees within 10 % with the PLASMA2 sun-photometer measured in the same environment.

The Single Scattering Albedo (SSA) is one of the most influencing parameters on the AAC DRE calculation that remains largely uncertain in models. During the AEROCLO-sA campaign, the average Single Scattering Albedo (SSA) obtained by OSIRIS at 550 nm is 0.87 in agreement within 3% on average with previous polarimetric-based satellite and airborne retrievals. The strong absorption of the biomass burning plumes in the visible is generally consistent with the observations from the Aerosol Robotic Network (AERONET) ground-based sun-photometers. This latter however shows a significant increase of the particles' absorption at 440 nm in the northern Namibia and Angola, which indicates more absorbing organic species within the observed smoke plumes. Biomass burning aerosols are also vertically collocated with significant amounts of water content up to the top of the plume around 6 km height in our measurements.

The detailed characterization of aerosol and cloud properties, water vapor and their uncertainties obtained from OSIRIS and PLASMA2 measurements allows to study their impacts on the aerosol above cloud DRE. The high absorbing load of AAC combined together with high cloud albedo leads to unprecedented DRE estimates, higher than previous satellite-based estimates. The average AAC DRE calculated from the airborne measurements in the visible range is +85 W m$^{-2}$ (standard deviation of 26 W m$^{-2}$) with instantaneous values up to +190 W m$^{-2}$ during intense events. These high DRE values, associated with their uncertainties, have to be considered as new upper

cases to evaluate the ability of models to reproduce the radiative impact of the aerosols over the South-East Atlantic
region.

## 1. Introduction

Aerosols from natural and anthropogenic sources directly impact the climate by interacting with solar and terrestrial radiations and indirectly through interactions with cloud properties (IPCC, 2013). According to their origin and the atmospheric transport, aerosol particles are unequally distributed in the troposphere where they can reside for several days or weeks. As a consequence, their chemical, optical and microphysical properties also present a strong variability (Lagzi et al., 2014).

Aerosol particles significantly impact the radiative budget of the Earth. However, due to the high variability of aerosol properties and their distributions in the atmosphere, the accurate quantification of their Direct Radiative Effects (DRE) remains uncertain. Climate models require several assumptions to represent aerosols, clouds and their interactions, resulting in a large diversity of aerosol DRE. The Aerosol Comparisons between Observations and Models (AEROCOM) experiment show biases between the most recent observations and models (Myhre et al. 2013; Samset et al., 2014). Significant biases are notably observed in particular in the South-Eastern Atlantic (SEA) region where highly absorbing particles co-exist with a low-level stratocumulus cloud (Zuidema et al., 2016).

The SEA region presents therefore a unique opportunity to study aerosol-cloud-radiation interactions and the impact of highly absorbing particles from biomass burning in central Africa, which are still debated (Bellouin et al., 2020). AeroCom study (Zuidema et al., 2016) demonstrates a net aerosol DRE from cooling to warming in this region. Indeed, Aerosol Above Cloud (AAC) highly contributes to climate uncertainties and very few methods currently allow the retrieval of a detailed model of their properties (Waquet et al., 2013b, Knobelspiesse et al., 2013, Peers et al., 2015). Peers et al. (2016) reported disagreements between five AeroCom models and satellite observations of AAC from the Polarization and Directionality of Earth Reflectances (POLDER) instrument. Most models do not reproduce the large aerosol load measured above clouds by POLDER. This study also demonstrated that large discrepancies exist between these climate models in terms of AAC absorption and load. The authors explained these differences in the models by the use of different parametrizations for the aerosol injection heights, vertical transport and absorption properties. De Graaf et al. (2014) demonstrated significant biases between the above cloud DRE estimated from satellite observations and modeled DRE in the SEA region. The modeled instantaneous DRE was estimated to be five times lower than the ones calculated with the measurements provided by the SCanning Imaging Absorption SpectroMeter for Atmospheric CHartographY (SCIAMACHY) sensor (i.e. 30-35 W m$^{-2}$ for SCIAMACHY retrievals versus 6 W.m$^{-2}$ for the DRE computed with a global model). Authors explained that these differences could be partially explained by an underestimation of the aerosol absorption in the visible and UV part of the spectrum. Mallet et al. (2020) also studied the sensitivity of the DRE to the absorption property of biomass burning aerosols in the SEA region using two regional climate models. Their works demonstrated a low bias in the modeled above cloud AOT of about 40% (in relative unit) between modeled and observed (Moderate-resolution Imaging Spectroradiometer, MODIS, and POLDER) above cloud AOD in the region which is necessarily reflected in an underestimate of the modeled DRE.

In order to retrieve aerosol DRE above cloud in the SEA region with well-defined uncertainties, which is needed to evaluate climate models, one need first to characterize aerosols and cloud optical and microphysical properties. Because the simultaneous retrieval of aerosol and cloud properties is still challenging (Cochrane et al., 2019), the European Space Agency (ESA) and the European Organisation for the Exploitation of Meteorological Satellites (EUMETSAT) developed a new spaceborne Multi-viewing, Multichannel, Multi-polarization Imager (3MI) to be launched in 2022 on-board the Meteorological Operational Satellite – second generation (MetOp-SG) satellite. To evaluate the next generation of retrieval algorithms, a 3MI airborne prototype, Observing System Including PolaRisation in the Solar Infrared Spectrum (OSIRIS, Auriol et al., 2008), has been developed at Laboratoire d'Optique Atmospheric (LOA, France). The total and polarized radiances sampled by OSIRIS between 440 and 2200 nm, and the new retrieval algorithms developed by Waquet et al. (2013) and Peers et al. (2015), allow to simultaneously retrieve the aerosol and the cloud properties in case of aerosols above clouds. Additionally, polarimetric measurements constitute a promising opportunity for the simultaneous retrieval of the aerosol and the surface properties (Dubovik et al., 2011, 2019).

In addition to aerosol-cloud interaction, the SEA region also represents a unique opportunity to study the direct radiative effects of highly absorbing particles transported above clouds. In this region, smoke aerosol plumes can

reach high altitudes (up to 6 km) and can be transported for several days in the atmosphere (Samset et al., 2014; Marenco et al., 2016) and overlay low-level clouds which are persistent over water. These considerations have motivated a significant number of intensive field campaigns between 1992 and 2018 (Formenti et al., 2019). Pistone et al. (2019) obtain Single Scattering Albedo (SSA) values for biomass burning aerosols from both airborne *in situ* and remote sensing methods during the Observations of Aerosols above Clouds and their Interactions (ORACLES) airborne campaign performed close to the Namibian coast in August-September 2016. From a sweeping view mode imager (the Airborne Multi-angle SpectroPolarimeter Imager, AirMSPI), mean SSA values at 550 nm were observed between 0.83 and 0.89 in August-September 2016. Mean SSAs ranging between 0.82 and 0.92 (from June to October 2006) were also retrieved at 550 nm over a large region centred on the South-east Atlantic Ocean using POLDER (Peers et al., 2016). A previous study based on Aerosol Robotic Network (AERONET) retrievals (Eck et al., 2013) has shown similar SSA values during the fire season. The latter study demonstrated that the seasonal trend of SSA in this region was mainly due to a change in aerosol composition, and particularly on the black carbon fraction.

Therefore, the new observation capabilities proposed by the airborne instrument OSIRIS give an interesting opportunity to characterise both cloud and absorbing particles in order to retrieve the aerosol DRE with high accuracy. Results are benefit to constrain climate models and satellite retrievals in a climate-sensitive region (Mallet et al., 2019).

In this paper we present aerosol and cloud retrievals performed over the SEA region essential for the calculation of the aerosol DRE. Measurements are performed by the OSIRIS and the PLASMA2 airborne instruments deployed during the AErosols, RadiatiOn and CLOuds in southern Africa (AEROCLO-sA) field campaign in Namibia during the biomass-burning period in 2017 (Formenti et al., 2019). Section 2 describes the flight trajectories and the main meteorological conditions encountered during the campaign. In Sect. 3, the OSIRIS, the airborne and ground-based sun-photometer and the airborne Lidar LNG retrieval methods are described. Section 4 reports the mean aerosol and cloud properties retrieved in the Namibian region essential for aerosol DRE retrievals. Finally, the results are summarised and discussed in Sect. 5.

## 2. Flights patterns and general atmospheric circulation description.

The AEROCLO-sA deployment comprised the measurements from the ground-based station of Henties Bay, Namibia (22°6'S, 14°30'E; 20 m above sea level (a.s.l.)) from 23 August to 12 September 2017 (Formenti et al., 2019).

The airborne component was conducted by the SAFIRE (Service des Avions Français Instrumentés pour la Recherche en Environnement) Falcon 20 aircraft from 5 to 12 September 2017. Ten flights were performed over ocean and land in the area from 7°30'E to 20°E and from 17°S to 22°30'S (**Fig. 1**). Two pre-campaign flights (not presented here) were also performed over the Mediterranean Sea during the summer 2017. Those flights correspond to pristine conditions over both clear and cloudy ocean scenes and were used for instrumental calibration.

The locations of the OSIRIS observations used in the present study are shown in **Fig. 1a**. Several filters are applied to the OSIRIS measurements to ensure optimum conditions for the retrieval (i.e. homogeneous clouds fields and stable flight conditions). As a first quality assurance, only stable flight conditions at high altitude (higher than 8 km a.s.l.) are selected. In addition, high altitude clouds and heterogeneous cloud scenes are rejected. The selected cases for OSIRIS inversions for aerosol above cloud represent 76% of cloudy measurements at high altitude (> 7 km). In these conditions, a total of 2h15m of OSIRIS measurements were processed. **Figure 1b** also represents the PLASMA2 measurements used in the present study which corresponds to low flight levels and stable flight conditions.

Data from the Copernicus Atmospheric Monitoring System (CAMS) reanalysis (Flemming et al., 2017) allow to obtain the biomass burning plume trajectory at 6-hour resolution at a 0.75°x0.75° spatial resolution. In **Fig. 2**, the atmospheric circulation and the associated biomass burning plume during the field campaign are represented by the geopotential height and wind at 700 hPa and the biomass burning Aerosol Optical Depth (AOD) at 550 nm. The plume path is displayed by highlighting the wind vectors at grid points with AOD higher than the 90[th] percentile of the AOD in the region (red arrows).

The regional atmospheric circulation on 8 September (**Fig. 2a**) represents the mean circulation during this period with air masses coming from tropical Africa, moving westward until Ascension Island which are then deflected to

the South-East due to the anticyclone centred over southern Africa. These conditions were observed during most of the AEROCLO-sA campaign except on 5 September (**Fig. 2b**), when the anticyclone was centred over the Indian Ocean between the South African coast and Madagascar. During this specific day, air masses were mostly transported over the continent and dust emissions were reported from both the climate model and the airborne lidar measurements (Formenti et al., 2019).

## 3. Instrumentation

### 3.1 The 3MI airborne prototype: OSIRIS

The OSIRIS imager provides both total and polarized radiances measurements. The airborne instrument is characterized by two optical systems: one for the visible and near infrared range (VIS-NIR, from 440 to 940 nm) with a wide field-of-view of 114° and one for the shortwave infrared (SWIR, from 940 to 2200 nm) with a field-of-view of 105°. The 2D detectors, which are respectively a CCD matrix of 1392x1040 pixels and a mercury cadmium telluride (MCT) focal plane array of 320x256 pixels allow to obtain very high-resolution images with a spatial resolution of 20 m for the VIS-NIR detector and around 60 m for the SWIR one at a height of 10 km. This high resolution allows to record the same scene up to 16 times (at 10 km height) from different viewing angles. Polarized measurements are available at 440, 490, 670, 870, 1020, 1600 and 2200 nm. Measurements without polarization capabilities are also performed in molecular absorption bands (763, 765, 910, 940, 1365 nm), and for a channel centred at 1240 nm, in addition to the channels previously listed.

The aerosol and cloud retrievals are performed using the OSIRIS measurements at 670 and 870 nm. A moving average is applied on measurements over a few pixels before the retrieval is achieved. The related radiometric noise is then estimated to be lower than $5.10^{-4}$ and $5.10^{-3}$ for the total and polarized normalized radiances, respectively. After all radiometric and geometric treatments are applied to the data (Auriol et al., 2008), the errors due to absolute calibration are expected to be lower than 3% for these channels. The absolute calibration accuracy was improved for the visible radiances using in-flight calibration technics (Hagolle et al., 1999) applied to OSIRIS measurements acquired over the Mediterranean Sea.

The algorithm used to retrieve the AAC properties with OSIRIS is based on an Optimal Estimation Method (OEM) developed for the POLDER instrument (Waquet et al., 2013). This method allows to simultaneously retrieve the aerosol and the cloud properties (Waquet et al., 2013 and Peers et al., 2015). Here, the aerosol retrieval is performed using the measurements in the solar plane of each image. The aerosol properties are then assumed to be spatially homogenous over the entire OSIRIS visible image (of about $20 \times 20$ km$^2$). Finally, the cloud properties are retrieved pixel by pixel over the entire image. As demonstrated in Waquet et al. (2013), this procedure increases the sensitivity of the algorithm to the aerosol properties.

The algorithm mainly provides AOD, SSA, extinction Angström exponent ($\alpha$) and the Cloud Optical Thickness (COT). SSA is defined as the ratio of the scattering to the extinction coefficient and primarily depends on the aerosol absorption (i.e. the imaginary part of the complex refractive index) and also the particles size (Dubovik et al.,1998; Redemann et al., 2001). The Angström exponent is indicative of the particles size (Reid et al., 1999; Schuster et al., 2006). The retrieved quantities are used to compute the instantaneous DRE over the solar spectrum, considering two main assumptions on the aerosol microphysics: (1) the complex refractive index of aerosols is assumed spectrally invariant; the imaginary part is retrieved and the real part value is fixed to 1.51, which is a reasonable average value for biomass burning aerosols (Dubovik et al., 2000) and (2) the particles size is retrieved only for the fine mode (particle diameters below 1 µm).. Moreover, the polarized measurements acquired for scattering angles larger than 130° are sensitive to the 3D cloud geometry effects on radiative transfer (Cornet et al., 2018). Since clouds are assumed to be plane parallel in the simulations, the method is simply applied to observations acquired for scattering angles smaller than 130°.

**Figure 3** shows an example of the measured and modelled radiances after the convergence is reached. It shows that the method allows to robustly model the selected data within the measurements noise. For these values of scattering angles, the sensitivity of polarization to cloud microphysics is minimized and the cloud droplet effective radius is assumed to be equal to 10 microns, which is the mean value for the stratocumulus clouds observed over the domain of interest (Deaconu et al., 2019).

### 3.2 The airborne Sun-photometer: PLASMA 2

PLASMA 2 (Photomètre Léger Aéroporté pour la surveillance des Masses d'Air version 2, Karol et al., 2013) is an airborne Sun-tracking photometer (referred to #950 on AERONET) which was on-board the SAFIRE Falcon 20 during the AEROCLO-sA campaign. The AOD of the atmospheric column above the aircraft is retrieved at nine wavelengths (340, 379, 440, 500, 532, 674, 871, 1020 and 1641 nm) from the PLASMA2 measurements. PLASMA (versions 1 and 2) observations have been validated against *in situ*, AERONET and satellite measurements (Mallet et al., 2016; Rivellini et al., 2017; Torres et al., 2017; Formenti et al., 2018; Hu et al., 2018), indicating that the accuracy on the AOD retrievals is of the order of 0.01 regardless of the wavelength. During AEROCLO-sA, several low-level flights were performed, typically near the cloud top when measurements were performed over the ocean and, near the ground, under clear sky conditions, when the low-level flights were performed over desert sites. PLASMA2 measurements performed at high altitude allowed to characterize the residual columnar AOD above the aircraft. This quantity was subtracted to PLASMA2 AODs measurements performed at low altitude for the sake of comparison with OSIRIS retrievals. Because PLASMA2 does not allow to perform almucantar measurements as AERONET Sun-photometers, the fractions of fine and coarse mode AOD was derived from the PLASMA2 spectral AOD measurements using the Generalized Retrieval of Atmosphere and Surface Properties (GRASP) algorithm (Dubovik et al., 2014). GRASP also allows to retrieve the volume size distribution from spectral AODs (Torres et al., 2017), assuming a complex refractive index (i.e. 1.50+0.025) and a bimodal lognormal particles size distribution with fixed modal widths.

### 3.3 Airborne lidar LNG

The vertical structure of the aerosol and cloud layers below the aircraft was obtained from the nadir-pointing airborne lidar LNG. The signal backscattered to the LNG system telescope at 1064 nm is range-square-corrected to produce atmospheric reflectivity. Total attenuated backscatter coefficient (ABC) profiles are derived from atmospheric reflectivity profiles by normalizing the atmospheric reflectivity above the aerosol layers to the molecular backscatter coefficient profiles. Hence the slope of the lidar reflectivity above 6.5 km a.s.l. matched that of the molecular backscatter derived from dropsonde measurements of pressure and temperature. The vertical resolution of the ABC profiles is 30 m and profiles are averaged over 5 s, yielding a horizontal resolution of 1 km for an aircraft flying at 200 m s$^{-1}$ on average. It is worth noting that ABC as observed with LNG is sensitive to both aerosol concentration and aerosol hygroscopicity. Indeed, relative humidity in excess of 60% modify the size and the complex refractive index of aerosol, and hence their optical properties, enhancing the ABC (e.g. Randriamiarisoa et al., 2006).

### 3.4 Ground-based AERONET sun-photometer measurements.

To put the aircraft observations into context, we analysed observations from 15 August to 15 September 2017 from the 15 AERONET stations located in the South-eastern Atlantic region. The full column integrated properties (i.e. AOD, complex refractive index, extinction Angström exponent, SSA and AOD fine mode fraction) have been averaged for each site and are shown in **Fig. 4.** Four AERONET stations are located in the AEROCLO-sA flight domain: Windpoort, Henties Bay, Gobabeb, as well as HESS, located in the southern part of the domain, 200 km south-East of Henties Bay, and outside of the flight tracks. During the campaign, and due to the persistent cloud cover, the measurements available at Henties Bay and Gobabeb were sparse. Therefore, the Windpoort station, located at 250 km from the Namibian Coast, is a more suitable station to monitor the aerosol evolution during the campaign. Additional interesting comparison data are provided by the Namibe station, located in the northern part of the AEROCLO-sA region, and more influenced by biomass burning emissions from central Africa than the Windpoort site (therefore exhibituing higher AODs). Outside the AEROCLO-sA domain, and as shown by the atmospheric circulation patterns in **Fig. 2**, biomass burning plumes were often transported towards Ascension Island. This remote location (i.e. 3000 km offshore the Angola coast) gives an opportunity to study the evolution of the biomass burning aerosols during their transport and aging (Zuidema et al., 2016; Mallet et al., 2019).

### 4. Results
#### 4.1 Aerosol extinction optical thickness.

The primary parameter influencing the aerosol above cloud DRE is the Aerosol Optical Depth (AOD) of the aerosol layer lofted above clouds. Above clouds AODs were measured directly with the sun-photometer PLASMA-2 during specific parts of the AEROCLO-sA flights. These high accurate AOD measurements allows for the validation of OSIRIS above clouds AODs as a first step of the study. AERONET measurements are also used in this section to depict the general AOD variability observed during the field campaign.

**Figure 5** reports the AERONET AODs measured at the Namibe, Windpoort, Ascension Island and Sao Tome sites during the biomass burning period (from 15 August to 15 September 2017). A moderate aerosol loading is observed at the Sao Tome and Namibe sites at the beginning of the selected period with mean AOD of 0.76 and 0.48 at 550 nm, respectively, whereas the Windpoort and Ascencion Island sites record mean AOD of 0.10 and 0.19 at 550 nm, respectively. A first increase of the aerosol loading is observed between 27 August and 1 September 2017 at the Namibe and Windpoort sites (AOD up to 0.88 and 0.73 at 550 nm, respectively), a second increase between 3 and 8 September at the Namibe, Windpoort and Sao Tome sites (AOD up to 1.80, 1.54 and 1.72 at 550 nm, respectively), and a third increase starting on 13 September at the Namibe and Windpoort sites (AOD up to 1.45 and 0.84 at 550 nm, respectively). During the AEROCLO-sA flight period, the strongest aerosol loadings in the AEROCLO-sA region were observed on 5 September and the lowest on 12 September from the AERONET dataset. This variation of the aerosol loading is mainly explained by the changes in the atmospheric circulation as demonstrated in Sect. 2.

The Aerosol Backscatter Coefficient (ABC) at 1064 nm on 12 September 2017 obtained from the airborne Lidar is shown **Fig. 6a**. The aerosol signal is mainly concentrated between the stratocumulus top at around 1 km and 6 km height. This vertical distribution well represents the general condition during the AEROCLO-sA campaign (Chazette et al., 2019). For this same day, **Fig. 6b** shows the spectral AOD measured by OSIRIS, PLASMA2 and the ground-based Windpoort and Namibe AERONET stations. The OSIRIS above-cloud retrievals were performed using measurements at an altitude of about 9 km a.s.l., corresponding to the top of the descent in loop. Contrary to the configuration of OSIRIS on the SAFIRE Falcon 20 aircraft, PLASMA2 is an upward looking instrument. Therefore, the above-cloud aerosol properties from PLASMA2 were obtained from the measurements at the bottom of the descent in loop, above the cloud top. AOD retrieved from OSIRIS and PLASMA2 measurements agree within ± 10% at 670 nm (**Fig. 6b**, **Table 1**), and are in between the measurements of the two AERONET stations. Bias between PLASMA2 and AERONET AODs are around 70% for every wavelength, whereas OSIRIS measurements agree with AERONET AODs from 20% with Windpoort measurements at 870 nm to 67% with Namibe measurements at 670 nm.

**Table 1** shows this same comparison from four loops, showing a very good agreement between OSIRIS and PLASMA2 for moderate to high aerosol loading (AODs from 0.36 and 0.74 at 670 nm). Note that, on 5 September, the aircraft did not reach the cloud top level at the end of the loop, which causes a higher bias (30%) between the AOD from PLASMA2 and OSIRIS's. OSIRIS AOD is slightly lower than the PLASMA one on 12 September by 0.04 for the AOT at 670 nm. This low bias can mainly be attributed to the neglect of coarse mode particles in the OSIRIS algorithm, which only considers fine mode particle to model the radiative properties of the aerosol biomass burning aerosols lofted above clouds. Indeed, on 12 September measurements, PLASMA2 inversion of the particle size distribution shows higher coarse mode particle concentrations (not shown) than for other flight measurements. This is probably due to some increase in the wind speed at the surface that uplifted some dust for this day and also because the concerned flight was performed straight along the Namibian coast. Thanks to PLASMA retrievals, we estimated the AAC coarse mode AOD to be equal to 0.04 at 670 nm during the loop performed above the clouds on 12 September. For the flights performed on the 7 and 8 of September, the departures observed between OSIRIS and PLASMA for the AOT at 670 nm are of about 0.01 (see table 1), which is the sun-photometer measurements accuracy. So, there is no systematic bias in the OSIRIS AOT retrievals. Based on the PLASMA and AERONET retrievals, we can affirm that the AAC coarse mode AOT can be safely neglected for the DRE calculations for all the flights of AEROCLO-sA, excepted on 12 September. The coarse mode AOD measured on the 12 of September limits the relative accuracy of our OSIRIS AOT retrievals at 670 nm to 10 %. For the calculation of the DRE uncertainties, we increased our AOT retrieval error to account for this observation (see Annexe B).

**Figure 7a** shows a time series of the OSIRIS above-cloud AOD at 490, 670 and 870 nm on 12 September 2017. The grey zones correspond to the OSIRIS retrieval uncertainty (as described in **Annexes A and B).** The 40-minute slot represents a transect of about 400 km along the coast-line (in yellow **Fig. 1a**) with a northbound heading. A slight variability is observed on the AOD with a south-north gradient. 30 minutes after the segment at high altitude, the collocated observations from PLASMA2 at low altitude are consistent with OSIRIS, with the AOD from PLASMA2 being larger by 10% than OSIRIS. Again, the low bias in the OSIRIS AOD retrievals (at 870 nm) is likely due to the presence of a few coarse mode particles, as previously discussed. Parallel measurements of the airborne lidar LNG demonstrate the small spatial variability observed on 12 September with homogeneous distribution of aerosols in the troposphere with low aerosol signal for the region and the season.

**Figures 7b** shows a histogram of the above-cloud AOD from OSIRIS at 550 nm for each flight of the campaign. The retrieved AODs are ranging from 0.2 to 1.2. The mean AOD depends on the flight with a maximum of 0.94

on 8 September and a minimum of 0.39 on 12 September. The 5, 7 and 9 September have a mean AOD values of 0.86, 0.73 and 0.60, respectively. These high AODs are consistent with the large value typically observed close to the coast by satellites (0.6 at 550 nm, Peers et al., 2016) and aircraft polarimeter measurements in the South-eastern Atlantic (around 0.75 at 550 nm, Pistone et al., 2019).

## 4.2 Angström exponent and particle size distribution

Aerosol size can have a significant impact on DRE calculation since it mainly controls the spectral dependency of the aerosol optical thickness. The Angström exponent is a parameter primarily indicative of the particles size. The Angström exponent retrieved with OSIRIS is evaluated against PLASMA measurements and particles size retrievals in this section.

**Figure 8a** describes the volume particle size distributions retrieved from PLASMA2 measurements at different altitudes during the descent in loop on 7 September 2017. The AERONET particle size retrievals from the Windpoort station are also shown. The size distribution is generally characterised by a dominant fine mode between 500 m and 5000 m. This is consistent with the dominant fine mode typically observed in previous studies at an altitude of 1 to 6 km in this region (Toledano et al., 2007; Russell et al., 2010; Kumar et al., 2013). Measurements show a rather constant fine-coarse mode ratios within the aerosol plume. For the descent in loop over cloud on 7 September 2017, fine mode particles contributed to 97 % of the total AOD at 670 nm between 1000 m and 4000 m. The mean Angström exponent value obtained from PLASMA2 measurements is about 1.9 with an accuracy of 0.1. According to AERONET measurements, during the campaign period, the amount of coarse mode particles is also extremely weak and does not exceed 5% of the columnar AOD (**Fig. 4**). The smallest particles (Angström exponent larger than 2) are generally observed in the northern part of the Namibian region (**Fig. 4**).

**Figures 8 b, c and d** show the altitude, the AOD and the extinction Angström exponent measured by PLASMA2 for straight levelled runs below 2000 m a.s.l.. The second flight on 12 September, over the Etosha Pan (black line) presents a strong AOD gradient between 0.30 and 0.82, and a strong Angström exponent gradient between 1.55 and 2.27, not correlated to altitude variations. During this specific flight, the measurements over the Etosha Pan were performed at around 1300 m a.s.l.. The Etosha Pan is the main source of dust emission in this region and it is characterised by an extremely dry surface at this time of year. In these conditions, the particles size distribution retrieved by PLASMA2 over the South-western part of the flight, over the Etosha Pan, is characterized by a significant additional contribution of coarse mode particles ($\alpha_{440-870} = 1.55$). Toward the North-Eastern part of the flight, the aerosol properties retrieved by PLASMA2 are less influenced by the dust emissions from the pan, and are more representative of the properties of fine mode particles from biomass burning ($\alpha_{440-870} = 2.27$).

The values of Angström exponents retrieved above clouds from OSIRIS and PLASMA2 are around 2.0 ± 0.2. In agreement with the PLASMA2 analysis, the $\alpha_{490-870}$ histogram shown in **Fig. 8e** indicates different aerosol types. The maximum $\alpha_{490-870}$ is observed on 12 September (median of 2.15), and the minimum on 8 September (median of 1.75). $\alpha_{670-870}$ values are generally constant during every flight, as in the case on 12 September (**Fig. 7a**). The lowest $\alpha_{490-870}$ values, around 1.7, are found less than 50 km away from the Namibian coast and were observed during the flight on 8 September. This behaviour might be explained by the influence of dust particles generated on the continent with higher coarse mode particle fraction. The hygroscopic growth of biomass-burning particles potentially occurring during their transport over the ocean might also explain the differences observed in the aerosol optical properties from one flight to another. The aerosols observed on 5 September were directly transported from Central Africa without a long transport over the Atlantic Ocean (**Fig. 2b**). This specific circulation might explain higher $\alpha_{490-870}$ (mean of 2.00) observed above clouds during this flight, which indicates smaller particles than the ones observed for instance on 8 September. These changes in the fine mode size are also suggested from the analysis of the volume fine mode properties obtained from PLASMA2 inversions. Lower values of the volume mean radius of the fine mode particles are retrieved on 5 and 12 September ($r_v = 0.18$ and 0.15 µm, respectively) rather than on 8 September ($r_v = 0.20$ µm).

AERONET Angström exponents are around 5 % lower than PLASMA2 measurements and from 8 to 25 % lower than OSIRIS inversions (**Table 1**). The bias between AERONET and OSIRIS Angström exponents can mainly be explained by the presence of coarse mode particles that are not explicitly considered in the POLDER algorithm when biomass burning layers are detected above clouds (Waquet et al., 2013a). This is particularly observed for the flight on 12 September due to a slightly less dominant fine mode compared to other flights (i.e. mean fine mode fraction of 90% on 12 September instead of 95% on average for the field campaign). This neglected coarse

mode could also explain why OSIRIS slightly underestimated AOD compared to PLASMA 2 on 12 September (**Fig. 7a**).

## 4.3 Complex refractive index and single scattering albedo

The SSA is one of the three most important parameters influencing aerosol DRE calculation with the above cloud AOT and the cloud albedo (Peers et al., 2015). The retrieval of this parameter is still subject to large uncertainties in this region (Peers et al., 2016, Pistone et al., 2019). The retrieval of the SSA from passive remote sensing technics depends on the microphysical assumption. This parameter is primarily driven by the aerosol absorption (i.e. imaginary part of the complex refractive index) and, to a lesser extent by the particles size. The analysis of AERONET measurements over the biomass-burning period also provides us with a spatially distributed view of the aerosol complex refractive index and single scattering albedo. Values of SSA at 675 nm (hereafter $SSA_{675}$) are generally spatially homogeneous over land with a mean value of 0.85 (25$^{th}$ and 75$^{th}$ percentiles of 0.84 and 0.86, respectively). The highest $SSA_{675}$ is observed at the Namibe station (**Fig. 4**) with a mean value of 0.87, while the lowest values are observed at the Bonanza and Mongu Inn stations (mean $SSA_{675}$ of 0.82). In correspondence, a mean refractive index of $1.51 + 0.027i$ at 675 nm is retrieved. The real part of the refractive index at 675 nm ranges from 1.41 at Henties Bay to 1.54 at Bonanza, and the imaginary part at 675 nm ranges from 0.008 at Henties Bay to 0.032 at Bonanza. The known environmental characteristics of Henties Bay, a coastal site with high content of sea salt and sulphate aerosols, frequent fog and a persistent as well as elevated relative humidity (Formenti et al., 2019; Klopper et al., 2020), support the low values of the real and imaginary refractive indices. **Figure 4** also shows different behaviours on the spectral variation of the imaginary part of the refractive index k in the South-Eastern region. $k_{441}$ is higher than $k_{675}$ (up to 43%) on the northern part of the region, and lower (up to 10%) on the southern part of the region. The largest ratio between $k_{441}$ and $k_{675}$ is observed at the Namibe station with $k_{441}$ higher than $k_{675}$ by more than 40%.

The SSA of AAC can also be observed from OSIRIS inversions for the entire AEROCLO-sA flight campaign. **Figure 9a** shows the SSA boxplots of each analysed AEROCLO-sA flight. Results show absorbing property in this region with a mean SSA of 0.87 at 550 nm for the full campaign. SSA values below 0.82 at 550 nm can be considered as outliers in our study. The highest and lowest absorption (and conversely the lowest and highest SSA) are retrieved respectively on 9 September (mean SSA of 0.86 at 550 nm) and on 5 September (mean SSA of 0.89 at 550 nm). This low variation in the measured particles absorption suggests rather uniform sources of BBA emissions, during the AEROCLO-sA campaign time scale.

**Figure 9b** compares the spectral variation of the SSA from OSIRIS during the full campaign to various complementary measurements: the concurrent ground-based AERONET at the Windpoort and Namibe sites, the retrievals of the airborne imager AirMSPI during ORACLES-2016 (Pistone et al., 2019), and the mean POLDER retrievals above cloud for the South-eastern region from 2005 to 2009 during the fire season (Peers et al., 2016). The SSA from OSIRIS are less than 1% different from the ORACLES-2016 AirMSPI observations. Both measurements are consistent with the multi-year average SSA from POLDER during the biomass burning period (less than 2% difference), as expected as these estimates are all based on polarimetric measurements. A higher bias (about 3%) compared to OSIRIS inversion is observed with AERONET retrievals at Windpoort, where the SSA is lower irrespectively of the wavelength. On the other hand, the measurements at the Namibe AERONET station are consistent with our retrievals at 670 nm, but not at 440 nm, where the SSA is significantly lower. This trend does not appear in the ORACLES measurements from AirMSPI, which were performed farther from the coast over the ocean. According to Kirchstetter et al. (2004), a decrease of the SSA at 440 nm can be partly explained by the presence of light-absorbing organic carbon (brown carbon). Unlike at the Namibe site, the SSA at Windpoort, located farther from the fire sources, was generally found to decrease from 440 to 875 nm.

## 4.4 Integrated water content

The biomass burning aerosol layers transported in the studied region are typically accompanied by water vapor, with potential significant effects on the radiative budget (Deaconu et al., 2019). It is therefore needed to consider the contribution of water vapour in our study in order to establish an accurate estimate of the aerosol DRE and related uncertainties. The integrated water vapour can be derived from the PLASMA extinction measurements performed at 940 nm (Halthore et al., 1997). In **Fig. 10**, one can note, in general, a linear relationship between the water vapour content and the AOD at 550 nm from PLASMA2, especially for the higher range of water vapour concentration. The highest column concentrations of water vapour (up to 2.4 g cm$^{-2}$) are observed for the two flights of 8 September. This correlation could be explained by the meteorological conditions, which would be responsible for the simultaneous transport of aerosols and water vapour, as suggested by previous studies (Adebiyi

et al., 2015, Deaconu et al., 2019). This correlation might be also the result of the direct emission of water vapour due to the fires themselves (Betts and Silva Dias, 2010; Sena et al., 2013). The water vapor amount is quite variable for one flight to another varying between 0.7 and 2.7 g.cm$^{-2}$. We estimated the mean water vapour amount to be equal to 1.7 g.cm$^{-2}$ for the AAC scenes sampled during AEROCLO-sA. Note that dropsonde measurements were used to supplement the PLASMA2 data in order to estimate the amount of water vapor within the cloud layer. Finally, one can note that there is no correlation for the second flight of 12 September between AODs and water vapor measurements. Low water vapour amount (below 1 g cm$^{-2}$) and high AOD values (>0.7) were observed together for this flight. We do not have full explanation for this contradictory observation. These observations were obtained for an in-land location (Etosha Pan) and we assume that this area could be associated with dryer air masses than the ones sampled over the oceanic regions. Based on aircraft measurements and model simulations in the South-Eastern Atlantic region, a recent study demonstrates that the water vapour concentration which is linearly correlated with CO concentration, may not originate to BB emissions (Pistone et al., 2021). Hence, the meteorology seems to mainly drive the amount of water vapour in the atmosphere in this region.

## 4.5 Cloud properties

The OSIRIS measurements also provide us with the optical properties of clouds with a high spatial resolution. For each pixel of the OSIRIS CCD matrix, the Cloud Optical Thickness (COT) can be retrieved at a spatial resolution reaching 20 m for a flight altitude around 10 000 m a.s.l.. **Figure 11a** shows a retrieved field of COT on 7 September at 09:37 UTC at the top of the first ascent of the flight close to the coast. COT values at 550 nm range from 5 to 30 with a mean value of 16.

To analyse the full AEROCLO-sA dataset, we chose to select the central pixel of every OSIRIS CCD matrix. This selection allows us to get rid of the scene details of each OSIRIS measurements and is well representative of the COT distribution in the region. As all the flights were performed between 8:00 and 11:00 local time, measurements refer to similar atmospheric thermodynamics and sun conditions. **Figure 11b** shows the distribution of COT values at 550 nm for the whole campaign. The COT at 550 nm ranges from 5 to 35. On 5 and 8 September, a mean COT of 12 is obtained while the mean COT is 15 on 9 September and 19 on 7 and 12 September. Particularly high values are observed on 7 and 12 September. As shown in **Fig. 11c**, COT can be related to the distance from the coastline. An increase from 10 to 30 is observed from the Namibian coast up to 100 km off the coast. On 7 September, measurements were performed up to 350 km away from the coastline. For this flight, a maximum value of COT is observed around 100 km away from the coast and then, the COT decreases down to 10 at around 250 km. It was noted, during the field campaign, that the clouds were generally optically less thick in the vicinity of the Namibian coast and more difficult to forecast.

## 4.6 Direct radiative effect

The Direct Radiative Effect (DRE) calculations are performed over the solar spectrum (0.2-4 microns) with the radiative transfer code GAME (Dubuisson et al., 1996). The Direct Radiative Effect (DRE) calculations are performed over the solar spectrum (0.2-4 microns) with the radiative transfer code GAME (Dubuisson et al., 1996). The DRE calculations are performed online and are based on OSIRIS retrievals performed in the visible and near-infrared spectral bands. The DRE for aerosols above clouds is primarily driven by the (spectral) AOT, SSA, and cloud albedo, this latter parameter being mainly controlled by the COT. The AAC DRE increases with increasing AOT and COT and decreasing SSA (or increasing aerosol absorption). To a lesser extent, the AAC DRE also depends on the water vapor and cloud droplets size. Increasing the amount of water vapor in the atmosphere tends to reduce the upwelling fluxes computed at the top of the atmosphere and consequently the DRE. Reducing the cloud droplets effective radius increases the cloud albedo, which increases the DRE. Our DRE uncertainty budget is based in the observational uncertainty previously determined for each aerosol parameter. Based on the comparisons made between OSIRIS and PLASMA, we estimated the uncertainties on the AOT and Ansgtröm exponent for the biomass burning aerosols (BBA) lofted above clouds to be respectively equal to 10% of the AOT at 550 nm and 0. 2 (for an Angström exponent computed between 670 and 865 nm). For the spectral SSA of the BBA, we considered a relative uncertainty of 5%, which encompasses the departures observed between the different airborne and spaceborne retrievals of this parameter (see figure 9). Considering the time period and location, a mid-latitude summer model was assumed to model the vertical profiles of the thermodynamical quantities (McClatchey, 1972). The total amount of columnar water vapor was fixed to a value of 1.7 g.cm$^{-2}$. We assume an error of ± 1 g.cm$^{-2}$ in accordance with the PLASMA observations for this quantity. We assumed a cloud droplets effective radius of 10 microns and perturbated this quantity by 2 microns in the calculations for the DRE uncertainty **(see annexe B)**.

The distribution of the observation-based, instantaneous and one-dimensional AAC DRE in the solar spectrum from OSIRIS measurements is shown in **Fig. 12** for every flight during AEROCLO-sA. Calculations show positive DRE in agreement with above-cloud observations from Zhang et al. (2016) in regions influenced by biomass burning emissions. Keil and Haywood (2003) also estimated a mean value of above-cloud DRE of +11.5 W m$^{-2}$ above cloud and of – 13 W m$^{-2}$ in clear sky conditions (mean SSA of 0.90 at 550 nm).

The above-cloud instantaneous DRE obtained from OSIRIS measurements ranges from + 10 to + 190 W m$^{-2}$ with mean values between + 65 W m$^{-2}$ on 12 September and +106 W m$^{-2}$ on 7 September. The mean DRE for the full campaign is +85 W m$^{-2}$ with a mean DRE uncertainty of 24 W m$^{-2}$. These mean DRE are higher than previous retrievals in the region. De Graaf et al. (2019a, 2019b) and Peers et al. (2015) retrieved a mean DRE around + 40 W.m$^{-2}$ from combined estimates made from satellite retrievals based on POLDER and SCIAMACHY sensors. In
these previous studies, the region of interest was larger (10°N-20°S, 10°W-20°E) than the one covered during the AEROCLO-sA aircraft campaign and these previous works may have included lower local values of DRE observed in the Western part of the region.

        These differences can be also explained by sampling issues due to the differences in the spatial resolution of the satellite and airborne sensors. Spatial resolution used by observations also have significant impacts on the
retrievals of cloud properties and consequently the estimate of the AAC DRE (De Graaf et al., 2019b). The so-called "plane-parallel bias" arises due to cloud sub-pixel heterogeneities and to the non-linear relationship of the radiance on the cloud optical thickness. For coarse-satellite data resolution (> 1 km$^2$), the known result is a low bias in the cloud optical thickness that increases as the spatial resolution of the sensor increases (Davis et al., 1997). This effect is necessarily reflected in an underestimation of the AAC DRE. For the spaceborne radiometers
POLDER and SCIAMACHY, the resolutions are respectively of 6 × 6 km$^2$ and 60 × 30 km$^2$. This is likely to explain why the POLDER DRE was estimated higher than the ones retrieved from SCIAMACHY (De Graaf et al., 2019b). The cloud fraction can also be overestimated for coarse-resolution satellite data, since the subpixel variability in the cloud properties cannot be sampled (Loeb and Davies, 1996, Di Girolamo and Davies, 1997). This effect also leads to lower estimates of cloud albedo and AAC DRE. We recall that the cloud optical
thicknesses retrieved with OSIRIS were performed at high spatial resolution (20 × 20 m$^2$) and often associated with full cloudy scenes, at the scale of the OSIRIS image (20 × 20 km$^2$). Altogether, these differences in the data sampling explains why the OSIRIS ACC DRE are larger than previous satellite-based estimates performed for this region.

        The exceptional atmospheric conditions sampled during the flights also largely explain why these retrieved high
DRE values. The high local DRE values observed during AEROCLO-sA are linked to the strong absorption properties and high aerosol loading typically measured during the field campaign. As demonstrated by Cochrane et al. (2019), based on the ORACLES campaigns in 2016 and 2017, the DRE also strongly depends on the cloud scene, in particular on the cloud albedo and cloud fraction. The stratocumulus cloud observed during AEROCLO-sA were generally optically thick (5 < COT < 30 at 550nm) and associated with geometrical cloud fraction of 1,
at the scale of the OSIRIS image. For instance, despite of the moderate aerosol loading observed on 7 September (mean AOD of 0.73 at 550 nm), the combination of absorbing particles and high cloud albedo (mean COT of 19 at 550 nm) leaded to the largest local DRE measured during the AEROCLO-sA field campaign (+106 W m$^{-2}$ ± 29 W.m$^{-2}$ on average). The relative low aerosol loading observed on 12 September (AOD of 0.12-0.18 at 865 nm) are still associated with significant values of DRE (+65 ± 25 W.m$^{-2}$ on average) mainly because of the high COT (20-
30 at 550nm) retrieved on 12 September.

## 5. Conclusion

        A new set of data of cloud and above-cloud aerosol properties allows to retrieve local aerosol above cloud DRE in the South-Eastern Atlantic region, where important biases persist between climate models for both the amplitude and the sign of the aerosol radiative perturbation. The detailed characterisation of the atmospheric particles content
achieved from the polarimetric imager OSIRIS allowed to study the sensitivity of the local radiative budget to the main optical aerosol and cloud parameters.

        Measurements were acquired during the AEROCLO-sA campaign over Namibia during the biomass-burning period of 2017 by the OSIRIS radiometer deployed during ten scientific flights from 5 to 12 September 2017. Five flights are selected for OSIRIS analyses presenting stable cloud conditions. Aerosol and cloud properties at a 20-
meter horizontal resolution with well-quantified uncertainties were retrieved from radiometric observations.

Measurements were performed in the highest range of retrieved AOD observed in this region and above a semi-permanent stratocumulus cloud deck. The high aerosol load (above-cloud AOD around 0.7 at 550 nm) retrieved from OSIRIS measurements matches the direct AOD measurements of the airborne Sun-photometer PLASMA2 within 0.05, or within 10% in relative unit, which contributes to validate the application of aerosol above cloud algorithms developed for POLDER (Waquet et al., 2020). Consistent (visible-near-infrared) Angström exponents values are retrieved from OSIRIS, , with values varying between 1.6 and 2.2, which is typical of BB aerosols, that are typically associated with submicronic particles. The PLASMA sun-photometer based retrievals suggest that changes in the fine mode particles size distribution are responsible for this variation in the Angström exponent. Ground-based AERONET measurements on the continent generally show slightly lower values of Angström exponent (close to 1.7 on average), which is likely due to the presence of additional coarse particles located in the boundary layer.

Biomass burning aerosols transported over the South-eastern Atlantic Ocean represent a high absorbing effect which significantly impacts the direct radiative effect. Mean above-cloud SSA of 0.87 at 550 nm were obtained from OSIRIS measurements. The SSA estimate is in an excellent agreement with previous (NASA) airborne and (CNES) spaceborne polarimetric retrievals performed over the same region (Peers et al., 2015, Pistone et al., 2019), given more confidence in the retrieval of this key parameter. Rather constant aerosol absorbing properties were observed in the Namibian region during the campaign in the visible range of OSIRIS. This suggests the BBA aerosols sampled were associated with rather homogenous sources at least at the time scale of the AEROCLO-sA airborne campaign. AERONET retrievals indicate an increase of the aerosol absorption at 440 nm over some specific sites, closer to the northern part of Namibia which were largely dominated by the contribution of fine mode particles indicating BBA aerosols. This result indicates a non-negligible concentration of UV light-absorbing organic matters (brown carbon) within the plumes, at least for the areas located closer to the fire regions.

As already noted by previous studies (Deaconu et al., 2019, Pistone et al., 2019), water vapour concentration and aerosol loading estimated above clouds are generally correlated in this region. These observations as confirmed by the sun-photometer measurements performed by the airborne PLASMA. So, our observations do not contradict previous studies indicating that both BBA aerosols and water vapor have to be considered together to investigate the total radiative impacts of smoke plume.

A significant part of climate uncertainty in the South-eastern Atlantic region is also explained by the lack of measurement of stratocumulus cloud properties. The characteristics of the airborne imager OSIRIS give the opportunity to retrieve cloud properties with a high spatial resolution. During the AEROCLO-sA campaign, a mean COT of 10 at 550 nm was observed. Measurements up to 350 km off the coastline allow us to probe stratocumulus property in different conditions. Results showed a maximum COT of 30 reached at 100 km off the coastline. As a result of the elevated above-cloud AODs, the strong absorption by aerosols, and the high cloud albedo, significant positive instantaneous aerosol direct radiative effects in the solar spectrum were observed close to the Namibian coast. The mean AEROCLO-sA instantaneous DRE value is +85 W m$^{-2}$ for AAC with mean uncertainties of $\pm$ 24 W.m$^{-2}$. We performed a detailed error budget for the DRE. Errors for the aerosol parameters (AOT, Angstrom exponent and SSA) were controlled based on comparisons of data from various sensors. This approach allowed a realistic calculation of the DRE uncertainties. This error budget also accounts for the variability observed for the water vapor during the flights and for the potential changes in the cloud particles microphysics. Our DRE estimates agree with previous studies indicating a strong positive aerosol forcing over the region (De Graaf et al., 2019b, 2020). Obtained DRE are generally higher than previous satellites ones, mainly because of the exceptional atmospheric conditions encountered during the flight (i.e. combination of high absorbing aerosol loads with high cloud albedo). As compared to previous satellite and modeled DRE obtained in the region, the airborne polarimeter used in the present study demonstrates high accuracy on the retrieved above cloud AOD, the absorption property and the cloud optical thickness in the visible-near-infrared domain. These well-defined aerosol and cloud properties have to be considered to evaluate the ability of models and satellites to reproduce locally high instantaneous DRE.

In conclusion, the airborne multi-viewing, multi-channel, multi-polarisation measurements in the region allow us to obtain aerosol and cloud properties with well characterized uncertainties as well as their sensitivity to aerosol above cloud DRE. Such findings are valuable to constrain climate models and also evaluate satellite retrievals as future 3MI measurements (Marbach et al., 2015). The high spatial resolutions, offered by the airborne polarimeter OSIRIS will allow to accurately estimate the DRE and the cloud properties and variability within regional model grids. Spectral extension of the OSIRIS algorithm will incorporate additional UV data from the airborne micro-polarimeter Ultra-Violet (MICROPOL), also operated on the Safire Falcon-20 during AEROCLO-sA, as well as

additional spectral bands in the middle-infrared (up to 2.2 microns). These will benefit to the characterization of the spectral absorption of the aerosol, linked to their chemical composition, and for the retrieval of the cloud microphysics, which is crucial for the study of the aerosol and cloud interactions. Lidar LNG profiles combined to OSIRIS data will allow to further evaluate the heating rate profiles of aerosol above clouds and to study the cases of interaction when aerosol and clouds are in contacts at the cloud top. Further investigation based on the

combination of this new set of observations and regional models, as described in Formenti et al. (2019), will be of greatly interest for such studies leaded in the SEA region.

**Acknowledgements**

The AEROCLO-sA project was supported by the French National Research Agency under grant agreement n°
ANR-15-CE01-0014-01, the French national program LEFE/INSU, the Programme national de Télédetection Spatiale (PNTS, http://www.insu.cnrs.fr/pnts), grant n° PNTS-2016-14, the French National Agency for Space Studies (CNES), and the South African National Research Foundation (NRF) under grant UID 105958. The research leading to these results has received funding from the European Union's 7th Framework Programme (FP7/2014-2018) under EUFAR2 contract n°312609″.

AC and FW acknowledge additional financial support provided by the Programme national de Télédétection Spatial (PNTS, grant n° PNTST-2020-06).

Airborne data was obtained using the aircraft managed by SAFIRE (www.safire.fr), the French facility for airborne research, an infrastructure of the French National Center for Scientific Research (CNRS), Météo-France and the French National Center for Space Studies (CNES). The AEROCLO-sA database is maintained by the French
national data center for atmospheric data and services AERIS.

The strong diplomatic assistance of the French Embassy in Namibia, the administrative support of the Service Partnership and Valorisation of the Regional Delegation of the Paris–Villejuif region of the CNRS, and the cooperation of the Namibian National Commission on Research, Science and Technology (NCRST) were invaluable to make the project happens.

FW also acknowledges the labex CaPPA, which supports his research. The CaPPA project (Chemical and Physical Properties of the Atmosphere) is funded by the French National Research Agency (ANR) through the PIA (Programme d'Investissement d'Avenir) under contract « ANR-11-LABX-0005-01 » and by the Regional Council « Hauts-de-France » and the « European Funds for Regional Economic Development » (FEDER).

**Author contributions**

AC performed simulations, analyses of airborne and AERONET data, and write the manuscript under the supervision of FW. PF, FW, CF, and MM, designed the original AEROCLO-sA observational concept, and co-led the 5-year investigation. FA, LB, CD, RL, and JMN developed, calibrated and assured high quality measurements of the OSIRIS instrument during the campaign. LB assured PLASMA2 calibration, settings and data processing. CD assured PLASMA2 measurements during the campaign. FW and FPe developed POLDER
algorithms. PG and FPa co-led the development of the OSIRIS and PLASMA2 instruments. OD and BT developed and allow access of the GRASP algorithm. MG analysed CAMS reanalysis data. CF provided LNG extinction profiles. Every co-authors contributed to the scientific analysis and to the writing of the manuscript.

**Competing interests**

PF is guest editor for the ACP Special Issue "New observations and related modelling studies of the aerosol–
cloud–climate system in the Southeast Atlantic and southern Africa regions". The remaining authors declare that they have no conflicts of interests.

**Data availability**

We thank the AERONET PIs Brent Holben, Jens Redemann, Carlos Ribeiro, Nichola Knox, Stuart Piketh, and their staff for establishing and maintaining the AERONET sites used in this investigation.

The AEROCLO-sA data are available in the BAOBAB platform available on https://baobab.sedoo.fr/AEROCLO/ and maintained by the French national data center Data Terra/AERIS. In particular, the dataset used in this paper are: OSIRIS (https://doi.org/10.6096/AEROCLO.1802), LNG (https://doi.org/10.6096/AEROCLO.1774), and PLASMA2 (https://doi.org/10.6096/AEROCLO.1807).

CAMS reanalysis data are available at the Copernicus Atmosphere Data Store,
https://ads.atmosphere.copernicus.eu/#!/home. Maps displaying the synoptic evolution of biomass burning aerosol

optical depth at 550 nm over Southern Africa and South Atlantic during the AEROCLO-sA campaign are available at https://baobab.sedoo.fr/Data-Search/?datsId=1782&project_name=AEROCLO.

Above cloud POLDER data called the AERO-AC products are available from the ICARE website: https://www.icare.univ-lille.fr/aero-ac/.

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

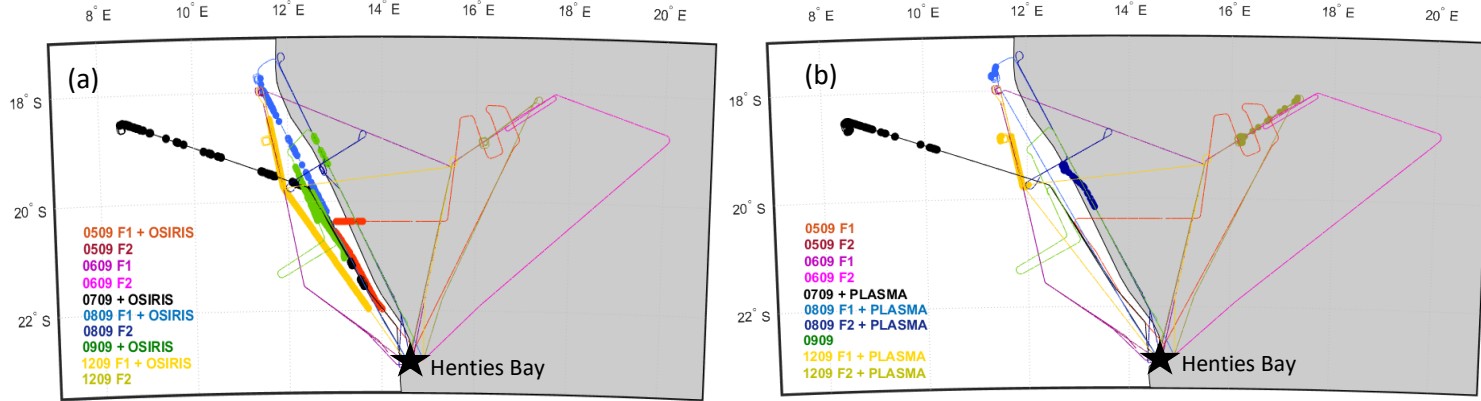

**Figure 1: Flight tracks of the 10 flights of the AEROCLO-sA campaign over the Namibian coast in September 2017. Circles indicate the (a) OSIRIS inversion locations and (b) PLASMA-2 measurement locations at low flight altitudes.**

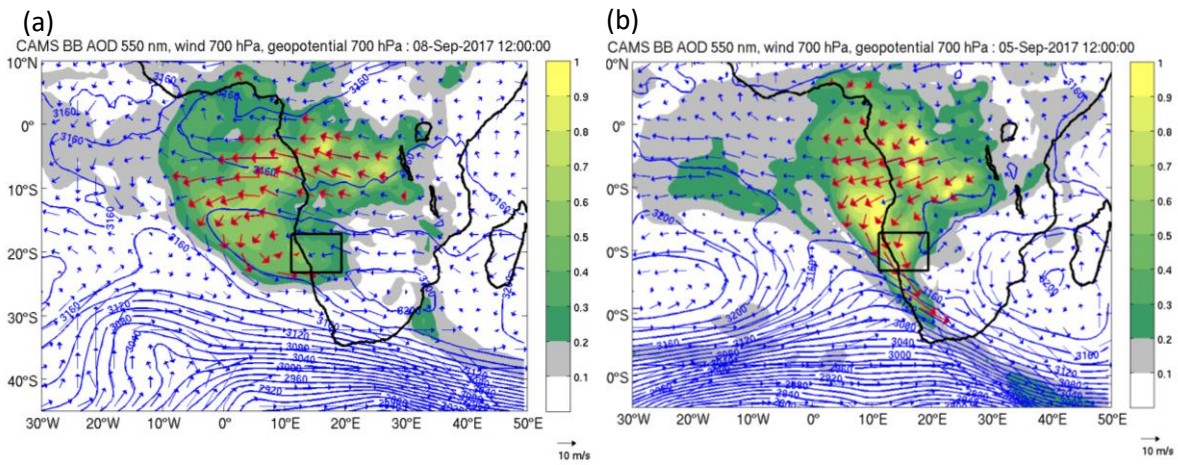

**Figure 2: Regional atmospheric circulation and aerosol for a) the 8th of September 2017 and b) the 5th of September 2017 at 12:00 UTC. Geopotential height (contours) and wind (arrows) at 700 hPa and biomass burning AOD at 550 nm (shadings) from CAMS reanalysis are displayed. Red arrows highlight wind vectors at grid points where the AOD is higher than the 90th percentile of the regional AOD. The AEROCLO-sA campaign region is located with a black rectangle in the Namibian region.**

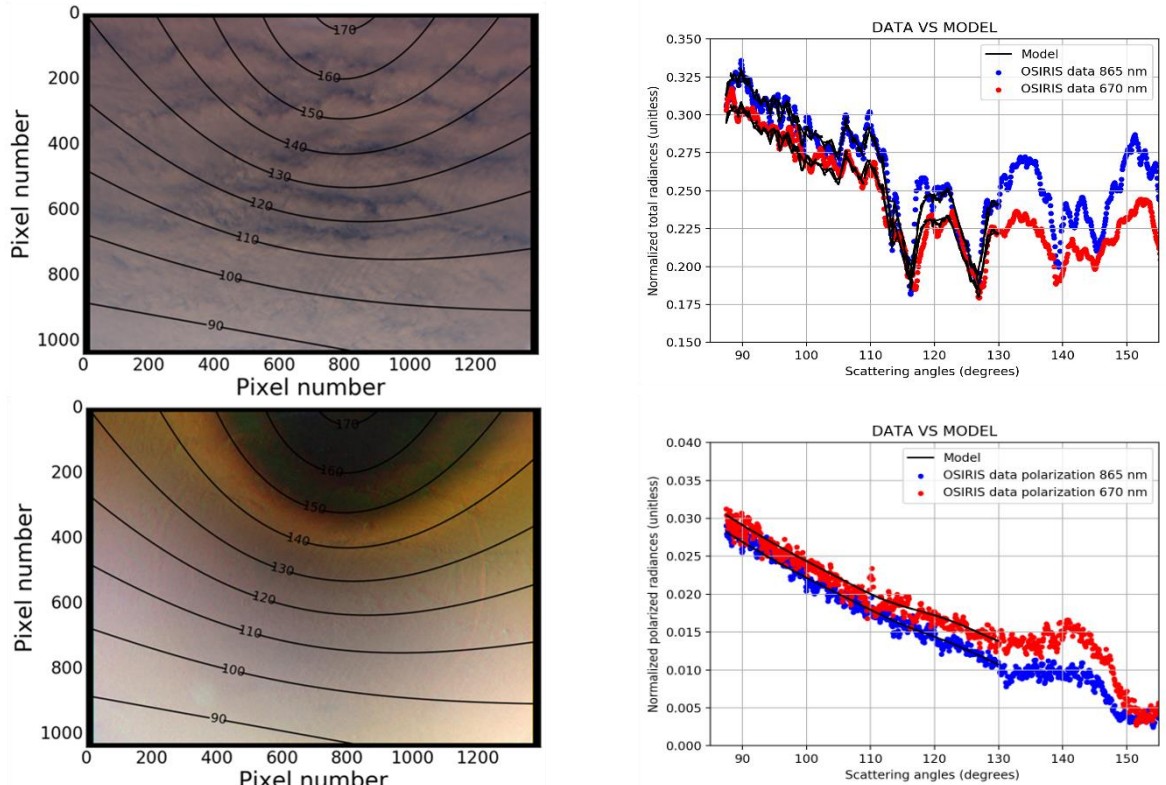

**Figure 3: Total and polarised radiances measured by OSIRIS during 8 September 2017. Left panels represent the total, on the top, and the polarised, on the bottom, recomposed RGB radiances for the full OSIRIS image. Right panels represent the corresponding principle planes at 670 nm (red) and 870 nm (blue). OEM's simulations according Waquet et al., (2013) of the total and polarised radiances for scattering angles below 130° are represented with black lines in the right panels. The main aerosol properties retrieved for this case are: AOD = 0.74 at 670 nm, $\alpha_{490-870}$ = 1.82, and SSA = 0.87 at 670 nm.**

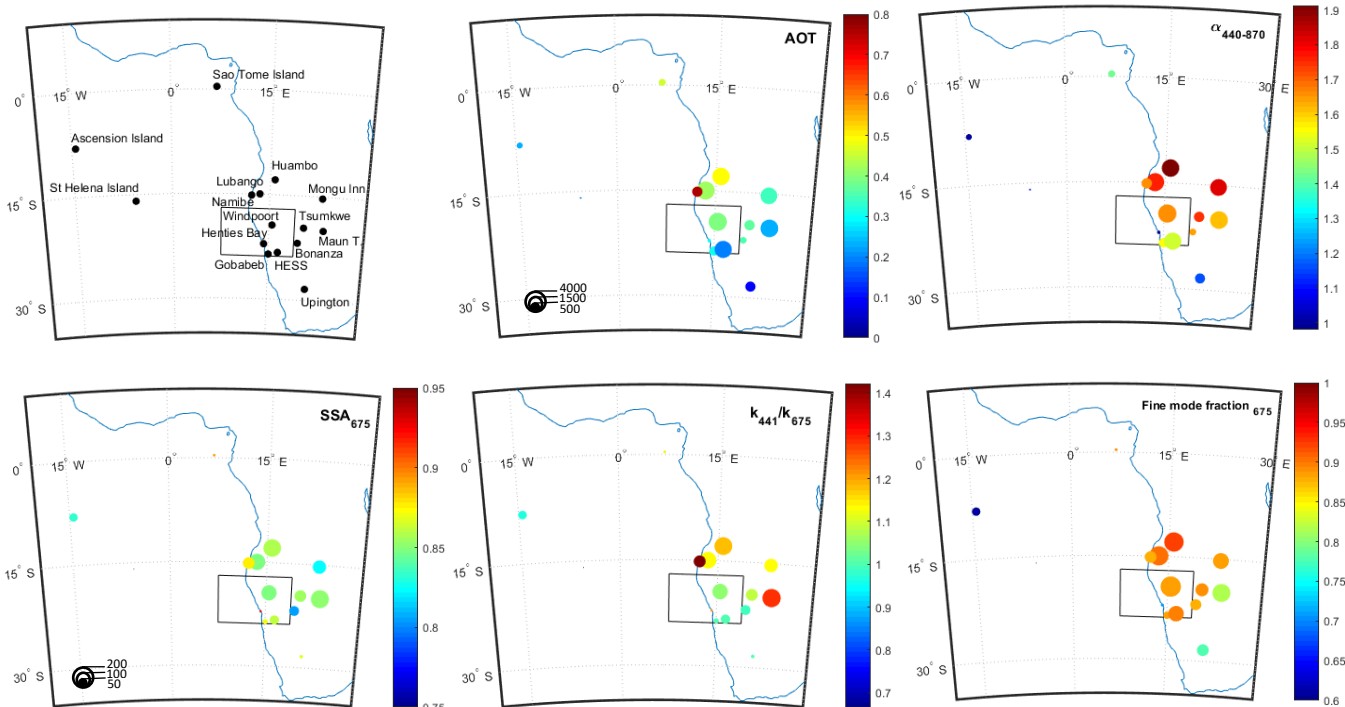

**Figure 4: Mean values of AOD at 550 nm, $\alpha_{440-870}$, SSA at 675 nm, ratio of the imaginary part of the refractive index (k) between 441 nm and 675 nm, and the fine to coarse mode fraction of the aerosol volume concentration from 15 AERONET sites in the South-Eastern Atlantic region. The selected period is from the 15 August to the 15 September 2017. The black box corresponds to the AEROCLO-sA flight domain. Circle size is linked to the availability of the data in number of measurements.**

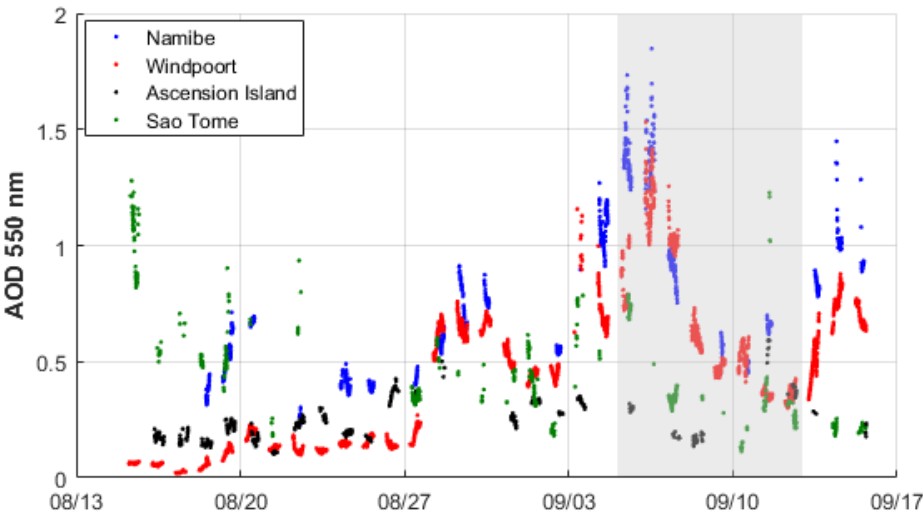

**Figure 5: AOD time series measured at 550 nm at Namibe, Windpoort, Ascension Island and Sao Tome AERONET sites from 13 August to 16 September 2017. The grey zone represents the flight period of the AEROCLO-sA campaign.**

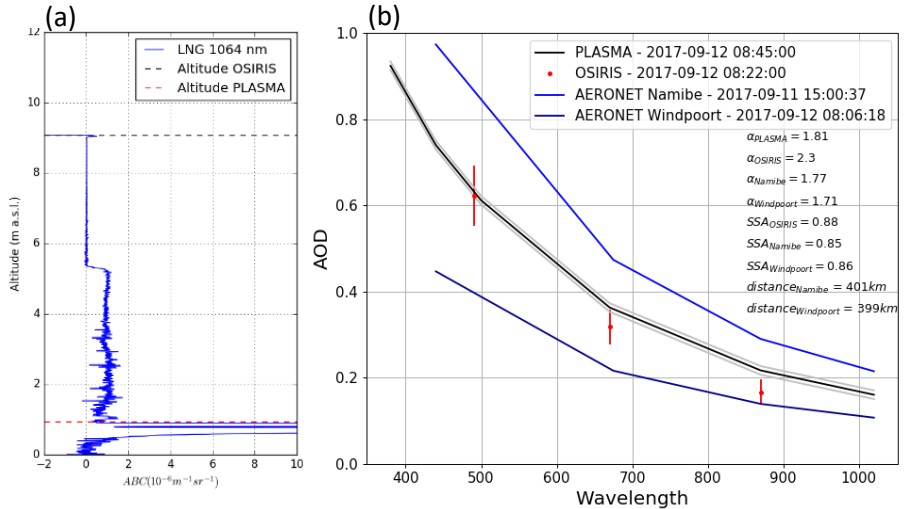

**Figure 6: a) Vertical profile of the Aerosol Backscatter Coefficient (ABC) measured at 1064 nm by the airborne lidar LNG before a loop descent performed on 12 September 2017. b) Wavelength dependences of AOD measured at Namibe and Windpoort AERONET ground-based stations, compared to OSIRIS and PLASMA-2 above clouds AODs, estimated before and after the descent in loop. Estimated error bars for AODs are also reported for OSIRIS (red bars) and PLASMA (grey zone).**

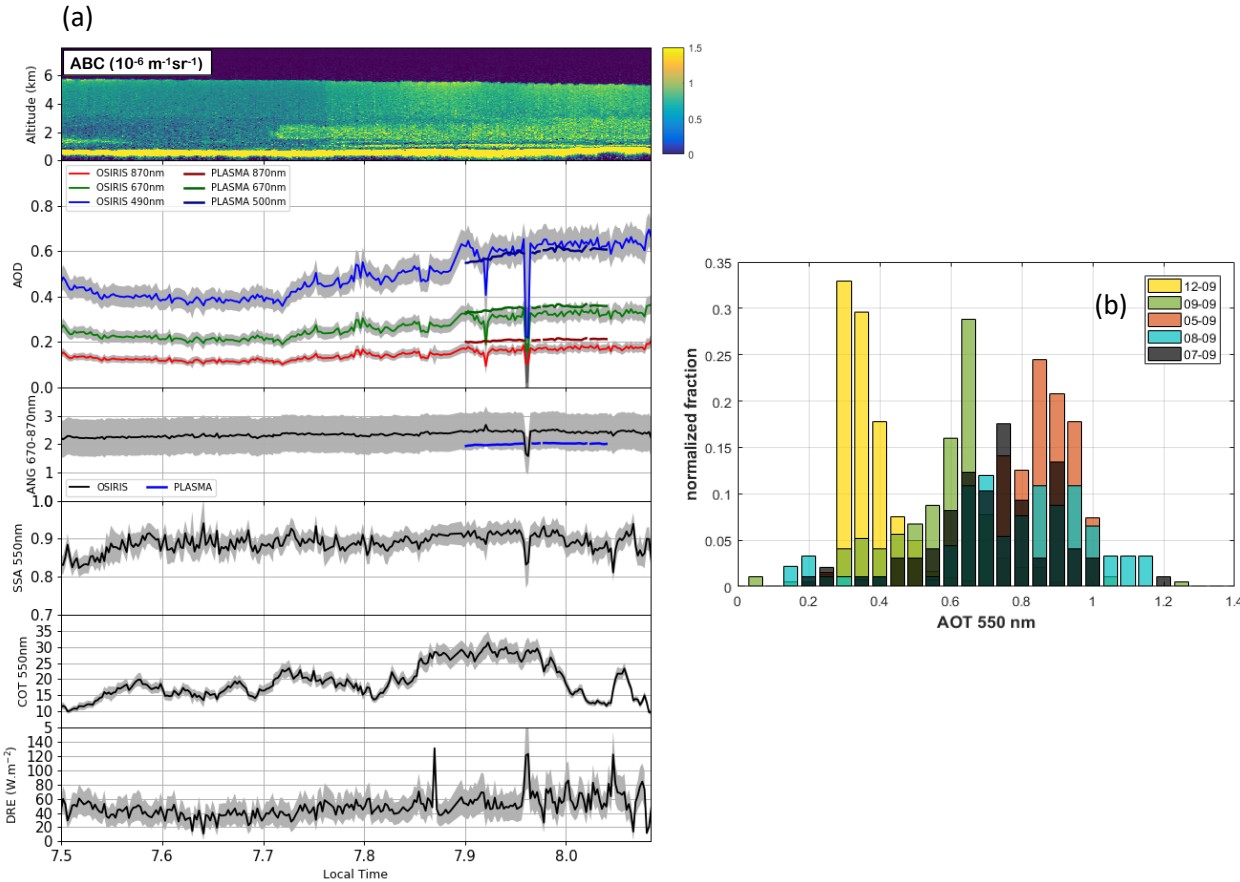

**Figure 7: (a) Time series of the ABC profiles measured at 1064 nm by the lidar LNG, above cloud AODs, Angström Exponents (670 nm - 870 nm), aerosol Single Scattering Albedo, Cloud Optical Depth and Direct Radiative Effect estimated on 12 September 2017, and associated errors. (b) AOD histograms from OSIRIS measurements for the five selected flights.**

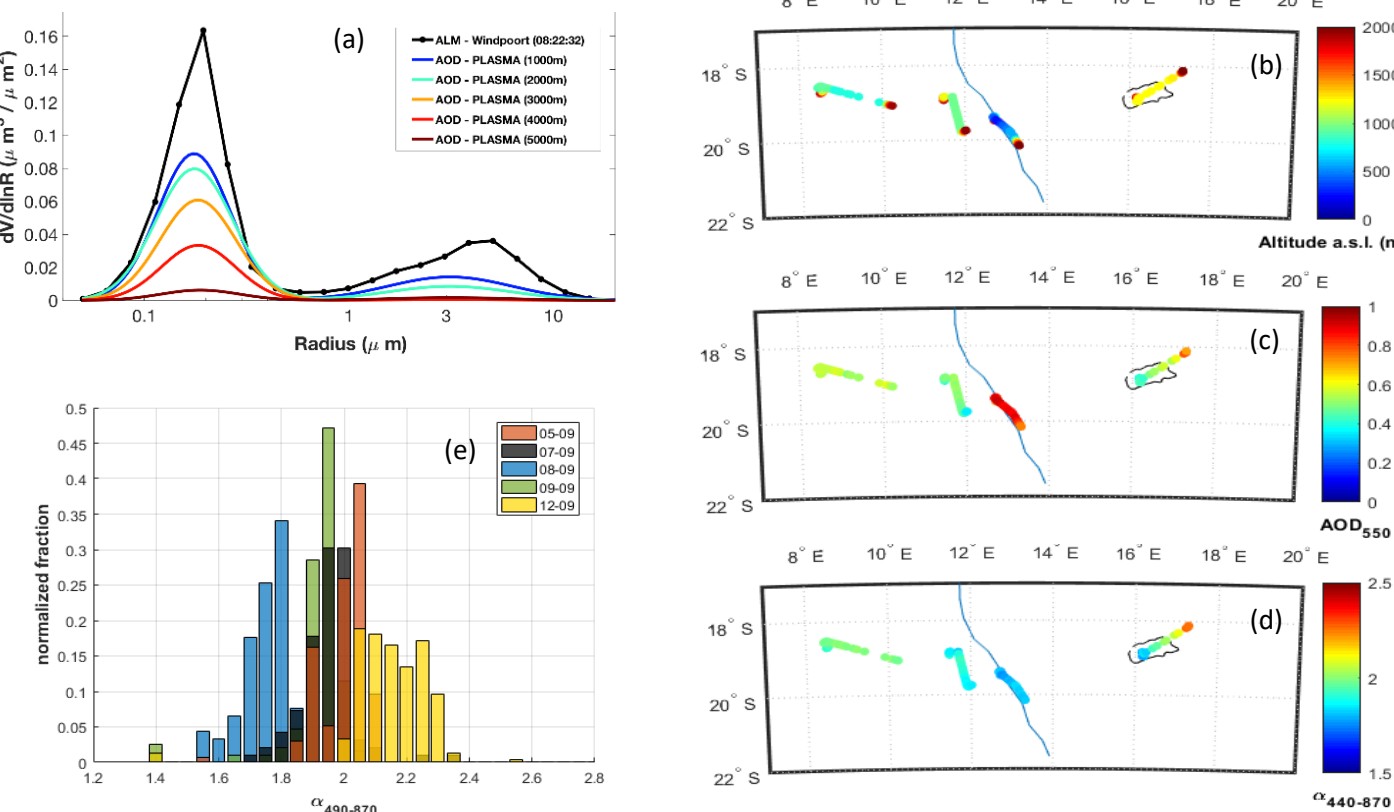

**Figure 8. (a) Retrieved volume particle size distributions at different altitudes from PLASMA2 measurements and from the AERONET station at Windpoort, Namibia, for the descent in loop of the 7 September 2017. Flight altitude (b), AOD (c) and Angström exponent (d) measured by PLASMA2 during AEROCLO-sA. Only low altitude flights are presented. The black line corresponds to the Etosha Pan. Only low altitude flights are presented. (e) Extinction Angström exponent from OSIRIS measurements histograms for the five selected flights.**

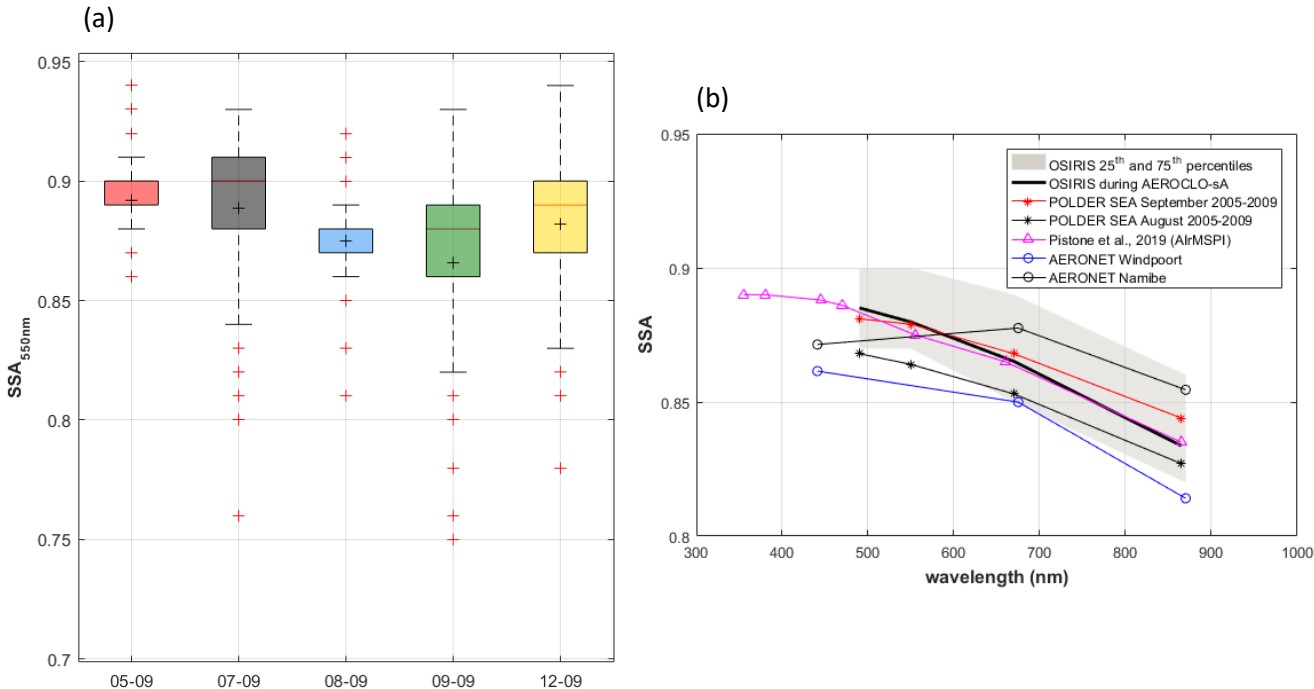

(a)

(b)

**Figure 9: Boxplot of the Single Scattering Albedo from OSIRIS measurements at 550 nm for the 5 selected flights and mean wavelength dependency retrieved from OSIRIS, Windpoort and Namibe AERONET sites during AEROCLO-sA campaign, AirMSPI during ORACLES campaign in 2016 and POLDER between 2005 and 2009 during the fire season.**

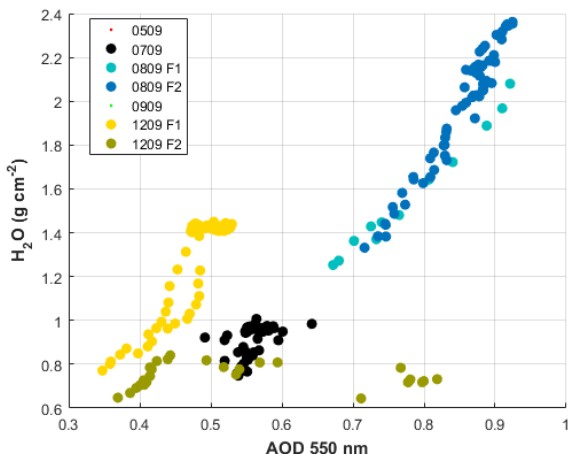

**Figure 10: Column water vapour in function of AOD at 550 nm both measured by the PLASMA2 Sun-photometer during the AEROCLO-sA campaign.**

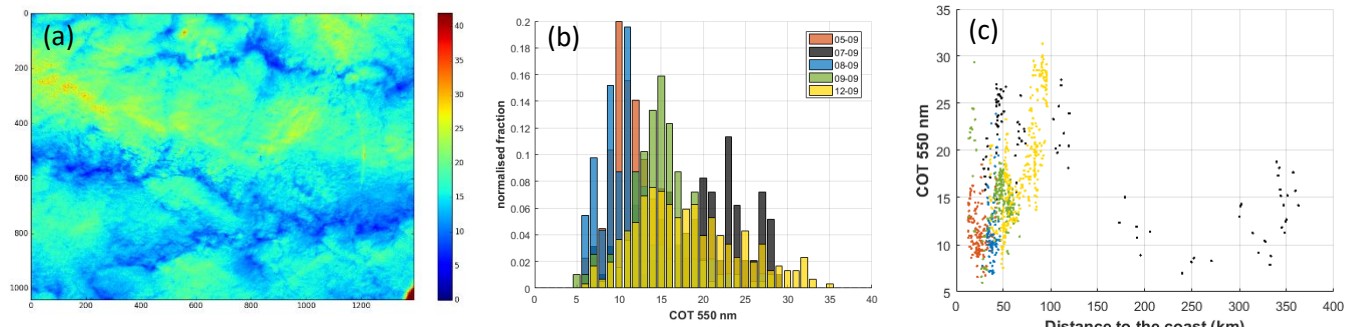

**Figure 11: a) Full OSIRIS image Cloud Optical Depth at 550 nm retrieved on 7 September at 09:37 UTC from OSIRIS measurements. b) COT histogram for the 5 selected flight. c) Relation between COT and the distance from the measurement to the coastline.**

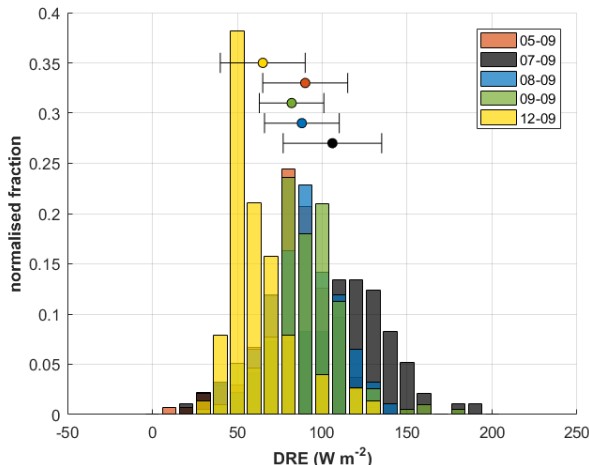

**Figure 12: Direct Radiative Effect (DRE) histogram retrieved from OSIRIS's aerosol and cloud retrievals for the 5 selected flights. Circles and error bars correspond to mean DRE and calculated DRE uncertainties following the description in Annexe B.**

**Table 1: Aerosol optical properties from AERONET, OSIRIS and PLASMA2 measurements for four different descents in loop during the AEROCLO-sA campaign.**


| Date | Altitude a.s.l. (m) min/max | $AOD_{670}$ OSIRIS | $AOD_{670}$ PLASMA | $AOD_{670}$ AERONET$_{Windpoort}$ and distance to F-20 | $\alpha_{490\text{-}870}$ OSIRIS | $\alpha_{440\text{-}870}$ PLASMA | $\alpha_{440\text{-}870}$ AERONET$_{Windpoort}$ |
|---|---|---|---|---|---|---|---|
| 05-09-2017 | 1244/9650 | 0.54 | 0.43 | 0.55 (373 km) | 2.06 | 1.82 | 1.76 |
| 07-09-2017 | 900/8680 | 0.44 | 0.45 | 0.75 (745 km) | 1.80 | 1.81 | 1.66 |
| 08-09-2017 | 686/9050 | 0.73 | 0.74 | 0.50 (464 km) | 1.80 | 1.69 | 1.66 |
| 12-09-2017 | 940/9080 | 0.32 | 0.36 | 0.22 (399 km) | 2.27 | 1.81 | 1.71 |

**ANNEXE A**

The Optimal Estimate Method (OEM) provides an error diagnostic of the retrieved parameters (Waquet et al., 2009). The square roots of the diagonal elements of the error retrieval covariance matrix $C_x$ gives the standard deviation associated with the retrieved parameters contained in the vector $X$. The OEM can be also used to estimate the standard deviation for other parameters that are non-included in this vector. For instance, for the single scattering albedo, the equation is given by :

$$\sigma_{SSA,\lambda} = \sqrt{\left( \sum_{i=1}^{N} \sum_{j=1}^{N} C_{x,i,j} \frac{\partial SSA_\lambda}{\partial X_i} \frac{\partial SSA_\lambda}{\partial X_j} \right)}$$

where the $N$ retrieved parameters contained in the vector $X$ are: the aerosol optical thickness, the imaginary part of the particles complex refractive index (aerosol absorption), the particles mean geometric radius and the cloud optical thickness.

**ANNEXE B**

For the direct aerosol radiative effect (DRE), the standard deviation is computed with the equation given by :

$$\sigma_{DRE} = \sqrt{\left( \sum_{i=1}^{N'} \left( \frac{\partial DRE}{\partial X_i} \right)^2 \sigma_{X_i}^2 + 2 \sum_{i=1}^{N-1} \sum_{j=i+1}^{N} \frac{\partial DRE}{\partial X_i} \frac{\partial DRE}{\partial X_j} C_{x,i,j} \right)}$$

where $\sigma_{Xi}$ correspond to the standard deviation associated with the parameters that control the DRE. The $\sigma_{Xi}$ terms related to aerosol properties were estimated based on observations to provide a more realistic error budget for the DRE. We considered a standard deviation of 10% to perturb the DRE calculations for the AOT at 550 nm (in relative value), a standard deviation of 0.005 for the aerosol absorption and a standard deviation of 0.02 microns for the particle's radius, corresponding to errors of 5% for the visible-near infrared SSA and 0.2 for the Angström exponent computed between 670 and 865 nm, respectively. Additional errors terms were also added to account for the variability in the water vapor amount (standard deviation of 1 g cm$^{-2}$) and for the cloud droplet effective radius (standard deviation of 2 microns). These last two errors terms were assumed uncorrelated and added to the single sum term appearing in the above equation.