# Peer review of "Aerosol above cloud direct radiative effect and properties in the Namibian region during AEROCLO-sA field campaign: 3MI airborne simulator and sun-photometer measurements."

_Atmospheric Chemistry and Physics, 2020_

## Referee Comment (RC1) · Anonymous Referee #1 · 6 Nov 2020

"Aerosol and cloud properties in the Namibian region during AEROCLO-sA field campaign: 3MI airborne simulator and sun-photometer measurements." by Chauvigné et al. presents the OSIRIS products such as aerosol optical depth and single scattering albedo obtained during the AEROCLO-sA airborne experiment. The presentation takes the form of histograms of individual variables, comparisons to reference observations and scatter plots between pairs of variables.

The paper, as it is, provides not so much science as retrieval results. Deeper analyses and discussions would be suitable. For example, no explanation is given for the

variability in the above-cloud SSA (between 0.75 and 0.95) (Line 331). What kinds of measured radiance spectra yield the low and high SSA values? Discussion is insufficient as to why the average SSA values, given only for two of the five flights, are judged similar to each other despite the 0.02 difference. It is also unclear in what logic the supposed similarity suggests the competing effects counteracting the meteorological variation. Doesn't the similarity rather suggest uniform source or retrieval inflexibility?

A more careful analysis for the AOD comparison would be nice too. In Line 246 the stratosphere is included in the PLASMA2 AOD but excluded from the OSIRIS. Can the stratospheric AOD value be estimated, for example from the PLASMA2 observations at the highest altitudes? Subtracting it from the low-altitude PLASMA2 observations may result in even better agreement with OSIRIS for the three days other than September 5.

Discussion on the apparent AOD-H2O relationship (Line 350) should address cloud contamination. Thin clouds can be mistaken as high AOD and are often coincident with high water vapor. How much of the data shown in Figure 10 were taken under clouds? How did you determine that?

As for the comparion of the calculated DRE values with other estimates, cloud fraction and intentional sampling bias should be mentioned in the place where the comparison is described (Section 4.6), without waiting until Lines 446 and 443. It is worth explaining and emphasizing why these two factors weaken the usefulness of the comparison.

Minor suggestions follow.

Line 3 Remove the period.

L. 16 Remove the first "of".

L. 22 Spell out AAC.

L. 25 Don't spell out AAC.

L. 27 "show" should precede "a".

L. 44 Replace telluric with terrestrial.

L. 95 "coexist" should read "coexists".

L. 105 Sect. 4 should read Sect. 5.

L. 119 Put "m" after 15.

L. 120 Remove the vertical bar.

L. 133 "the" should precede "airborne".

L. 140 "wild" should read "wide".

L. 149 Did you mean to put "a" before "few"?

L. 184 Put "to" after "referred".

L. 190 "overcoast" should read "overcast".

L. 261 The backscattering of 1.10-6 m-1sr-1 is not necessarily low. Shinozuka et al. (2020) used a threshold one fourth as large to identify smoke. How about noting the value under which 95% of the Sept 12 data reside, instead of 100%?

L. 346 Drop s from "coasts".

L. 355 End the sentence with a period after "area". Capitaliza b in "biomass".

L. 406 Insert "horizontal" before "resolution".

L. 411 Put an s to "contribute".

---

## Referee Comment (RC2) · Anonymous Referee #2 · 25 Nov 2020

In this study, the authors document imager and sun-photometer retrievals of above-cloud aerosol depth, single-scattering albedo, Angstrom exponent, refractive index, and size distribution made during the AEROCLO-sA aircraft field campaign over the south-eastern Atlantic. Retrievals are compared to similar measurements by similar aircraft instruments and ground-based sun-photometers. Estimates of cloud optical depth and column water vapour are also discussed.

The paper is straightforward, discussing each variable in turn with a few supporting figures, with brief, often speculative, explanations for differences between flights and

instruments. The paper would therefore be a useful citation for future users of the data. But the paper reads too much like a data description document and does not align with the ACP scope of publishing "studies with general implications for atmospheric science". However, I think it is possible for the paper to actually tell a story that would make it a valuable contribution to ACP. I also find that several conclusions are insufficiently supported by the analysis. For these reasons, which are detailed below, I recommend major revisions to improve the discussion.

**1  Main comments**

- The paper covers a succession of aerosol and cloud variables, without a clear scientific question to answer. That makes it a frustrating read. One solution would be to make the calculation of the direct radiative effect (section 4.6) the objective of the paper. To do that calculation, one needs to characterise aerosols, clouds, etc., which motivates the need for sections 4.1 to 4.5. Note that an analysis of uncertainties in direct radiative effect, propagated from the retrieved optical properties, would be required.

- The paper opens with a tantalising description of a new instrument, 3MI (lines 83-92), and concludes that 3MI improves the "definition" (unclear what is meant by that word) of above-cloud aerosol properties (lines 449-451). But the discussion does not clearly identify what new abilities 3MI brings compared to its predecessor, POLDER, and how the AEROCLO-sA field campaign helps demonstrate these new capabilities. The discussion needs to support that conclusion. What did the authors do that could not be done before? What has been done better?

- The abstract reads that "Combination between water vapour and the strong positive aerosol forcing over the region explains possible feedbacks on cloud development." (lines 40-41) What does that mean? Which part of the discussion

supports that statement?

- The authors find a consistent 10% disagreement between aerosol optical depth from OSIRIS and PLASMA (lines 246, 259, 411). But the implications of that disagreement are never discussed. Is that something to worry about? Is that a systematic bias? What causes it?

**2  Other comments**

- Lines 49-50: Need a reference for that statement. Note that the conclusions of Bond et al. (2013 doi:10.1002/jgrd.50171) have been challenged, see the discussion in section 5 of Bellouin et al. (2020 doi: 10.1029/2019RG000660)

- Line 117: I would not call a 35% reduction in data amount slight!

- Lines 177-181 and Figure 3: I understand that Figure 3 is there to illustrate the retrieval process, but it would be useful to have an idea of the outcome. Which "model" fits the data best? With what optical parameters?

- Lines 184 and 191: Does PLASMA2 onboard the aircraft performs the almucantar scans used by the AERONET inversion algorithm? Or are the size distributions derived in other ways?

- Lines 189-191: This statement is confusing. What do the authors mean by "low-level flight" in this context? Near the ocean surface or near the cloud top?

- Line 252: "robust independently on the aerosol loading." Too strong a statement considering the limited range of aerosol optical depth shown in Table 1.

- Line 355: What do the authors mean by "wood moisture"? The moisture emitted by evapotranspiration of forests?

- Line 359: "This particularity" – what does it refer to? The absence of correlation, or the low water content? It is unclear whether the different behaviour seen in Figure 10 for flight 1209 F2 is in fact understood.

- Line 383: " DRE calculations are performed online". What does that mean?

- Line 408: " extreme environment". In what sense? Having a large aerosol loading probably makes retrieving their optical properties easier.

- Line 419: " significantly impacts the climate". This conclusion is outside the scope of the study, so should either be supported by references, or made conditional.

**3 Technical comments**

- Line 17: analyse of -> analyse

- Abstract: Some acronyms are defined, but others (OSIRIS, PLASMA2, POLDER) are not. Need to make it consistent.

- Line 27: a show -> show a

- Line 44: "telluric": English speakers tend to prefer "terrestrial"

- Line 80: change on -> change in

- Figure 1: the coastline could easily be mistaken for a flight track! Perhaps set a Google Earth background, or blue over ocean, or something like that?

- Line 133: airborne the -> airborne

- Line 190: overcoast -> overcast

[Figure]

- Line 209: form -> form

- Line 256: north cape -> northbound heading

- Caption of Figure 7: need to state that panel b is from OSIRIS measurements.

- Line 315: spatial -> spatially

---

## Author Comment (AC1) · 26 Feb 2021

First, authors would like to thank the reviewer for his/her interesting suggestions. We believe they clearly helped to bring important details on the analyse, to clarify different aspects of our findings and to better fit with the ACP scope. We did our best to improve the paper and to respond to the reviewer questions.

In the following, authors answer to the reviewer and list modifications made to the paper.

"Aerosol and cloud properties in the Namibian region during AEROCLO-sA field campaign: 3MI airborne simulator and sun-photometer measurements." by Chauvigné et al. presents the OSIRIS products such as aerosol optical depth and single scattering albedo obtained during the AEROCLO-sA airborne experiment. The presentation takes the form of histograms of individual variables, comparisons to reference observations and scatter plots between pairs of variables.

The paper, as it is, provides not so much science as retrieval results. Deeper analyses and discussions would be suitable. For example, no explanation is given for the variability in the above-cloud SSA (between 0.75 and 0.95) (Line 331). What kinds of measured radiance spectra yield the low and high SSA values? Discussion is insufficient as to why the average SSA values, given only for two of the five flights, are judged similar to each other despite the 0.02 difference. It is also unclear in what logic the supposed similarity suggests the competing effects counteracting the meteorological variation. Doesn't the similarity rather suggest uniform source or retrieval inflexibility?

**Main answer:**

As also recommended by the reviewer 2, the text has been modified in order to highlight the main scientific goal of the study, which is now the quantification of the direct radiative effect of above cloud aerosol (AAC DRE) over the SEA region. This is a major source of uncertainties between the climate models (Stier et al., 2013) and, so, the estimate of the AAC DRE is one of the main scientific questions that the researchers involved in the ORACLES, CLARIFY and AEROCLO-sA field campaigns try to solve. The main structure of the manuscript remains unchanged since the complete characterisation of aerosol and cloud properties is necessary to estimate the DRE. At this occasion, uncertainties on the above cloud DRE have been evaluated and added. For the aerosol parameters, the DRE errors budget is based on the observational uncertainties obtained from the comparisons performed between the different sensor included in AEROCLO-sA and from other previous observations provided for instance during the ORACLES field campaign. We think that this approach allowed to implement a realistic and rigorous DRE errors budget. A sensitivity study for the DRE and additional discussions were also included. In addition, as recommended by the reviewer 1, the discussion on the absorption property retrieved from OSIRIS has been clarified.

Here below, we investigate the different points of the reviewer's comment.

For example, no explanation is given for the variability in the above-cloud SSA (between 0.75 and 0.95) (Line 331).

**Answer:**

First, we agree that the analysis of our results obtained for the aerosol single scattering albedo (SSA) was not clear enough. Histogram of the SSA has been replaced by boxplot showing percentiles, median and mean values for each flight as shown by **Figure RC1.1**. The figure shows that the range of above-cloud SSA is in fact narrower than previously expected. Values of SSA smaller than 0.82 and larger than 0.93 are not statically representative and can be considered as outliers. Airborne remote sensing of aerosol properties remains an experimental work and this is not surprizing that a tiny percentage of our retrievals may have not reached the expected quality despite of the use of quality filters.

[Figure]

**Figure RC1.1: Boxplot of the Single Scattering Albedo from OSIRIS measurements at 550 nm for the 5 selected flights.**

**Modifications:**

l.384: "SSA values below 0.82 at 550 nm can be considered as outliers in our study. The highest and lowest absorption (and conversely the lowest and highest SSA) are retrieved respectively on 9 September (mean SSA of 0.86 at 550 nm) and on 5 September (mean SSA of 0.89 at 550 nm). This low variation in the retrieved SSA of particles suggests rather uniform sources of BBA emissions, during the AEROCLO-sA campaign time scale."

l.534: "This suggests the BBA aerosols sampled were associated with rather homogenous sources at least at the time scale of the AEROCLO-sA airborne campaign"

Figure 9a has been changed to boxplot format.

What kinds of measured radiance spectra yield the low and high SSA values?

**Answer:**

The spectra observed for both polarized and total radiance are multiple and complex for AAC scenes and does not only depend on the SSA value. For the total radiance, and for a fixed value of above cloud scattering optical thickness, increasing the imaginary part of the complex refractive will reduce the SSA and consequently the absolute value of the total radiances (attenuation effect). This effect will also increase the relative differences between the measured spectral radiances (the so-called "color effect" linked to the change in the apparent color of the cloud due to increasing BBA absorption). However, this latter effect largely depends on the brightness of the below cloud layer. Different combination of COT, AOT and SSA may lead to rather similar spectra for the total radiance.

The measured radiance spectra is not the only physical effect that controls the retrieval of the above-cloud SSA in our algorithm. The retrieval of the above-cloud SSA depends also largely depend on the spectral polarized radiances and on their directional behaviours (see Figure 2 in the manuscript). Polarization measurements are mainly sensitive to the scattering aerosol optical thickness and to the change in the fine mode particles size, which also controls the SSA.

Discussion is insufficient as to why the average SSA values, given only for two of the five flights, are judged similar to each other despite the 0.02 difference.

**Answer:**

A change of 0.02 in the SSA is not significant since it is below our retrieval's uncertainties for this parameter. We recall that the uncertainty associated with the AERONET sun photometer retrievals cannot exceed 0.03 for the SSA. The retrieval of the SSA is very difficult but the estimate of the uncertainties associated with this parameter also remains a difficult task. This is mainly because this parameter cannot be directly measured and its estimate requires assumptions.

Inter-comparisons of various SSA estimates is probably the best way to currently reduce the variability associated with this parameter. As in Pistone et al., 2019 (ACP), we performed an inter-comparison of SSA retrievals performed from multiple instruments: airborne polarimeter from NASA (airMSPI), LOA (OSIRIS) and CNES spatial polarimeter POLDER. An excellent agreement of 3% was found between these different estimates that brings confidence in the evaluation of this highly uncertain and key parameter at least for the region and time period considered in the present study.

We added a sentence in the abstract to underline this important result, which is obtained by the inter-comparison of data provided by different instruments, teams and field campaigns.

**Modifications:**

l.34: "During the AEROCLO-sA campaign, the average Single Scattering Albedo (SSA) obtained by OSIRIS at 550 nm is 0.87 in agreement within 3% on average with previous polarimetric-based satellite and airborne retrievals.".

It is also unclear in what logic the supposed similarity suggests the competing effects counteracting the meteorological variation.

**Answer:**

We agree with the reviewer. Our sentence was unclear and misleading. Considering the small variability observed in our above-cloud SSA retrievals, and considering also that the time duration of the AEROCLO-sA airborne field campaign was only one week, we now conclude that our observations indeed rather suggest a uniformity of the sources of BBA emissions, at least at the scale of the airborne AEROCLO-sA field campaign.

**Modifications:**

l.387: "This low variation in the measured particles absorption suggests rather uniform sources of BBA emissions, during the AEROCLO-sA campaign time scale."

Doesn't the similarity rather suggest uniform source or retrieval inflexibility?

**Answer:**

We want to make clear that this is not an issue of retrieval inflexibility.

A similar methodology as the one used in the present paper was previously applied to the data provided by the POLDER spaceborne sensor (Peers et al., 2015). The method was able to distinguish between strongly absorbing biomass burning aerosols, boreal biomass burning aerosols, with rather scattering properties, and strong events of mineral dust above clouds transported from the Sahara. The method allowed to estimate different particles size and SSA values ranging between 1 and 0.8, which clearly shows that the retrieval technic is flexible and sensitive to the changes in the aerosol properties.

A more careful analysis for the AOD comparison would be nice too. In Line 246 the stratosphere is included in the PLASMA2 AOD but excluded from the OSIRIS. Can the stratospheric AOD value be estimated, for example from the PLASMA2 observations at the highest altitudes? Subtracting it from the low-altitude PLASMA2 observations may result in even better agreement with OSIRIS for the three days other than September 5.

**Answer:**

Indeed, PLASMA2 measurements includes the whole atmospheric column located above the aircraft. Possible high-altitude aerosols are thus observed from PLASMA2 measurements. However, this has already been

considered in our comparison by subtracting PLASMA2 measurements at high altitude to PLASMA2 measurements at low altitude for the comparison. This point has been clarified in the text.

**Modification:**

l.211: "PLASMA2 measurements performed at high altitude allowed to characterize the residual columnar AOD above the aircraft. This quantity was subtracted to PLASMA2 AODs measurements performed at low altitude for the sake of comparison with OSIRIS retrievals. Because PLASMA2 does not allow to perform almucantar measurements as AERONET Sun-photometers, the fractions of fine and coarse mode AOD was derived from the PLASMA2 spectral AOD measurements using the Generalized Retrieval of Atmosphere and Surface Properties (GRASP) algorithm (Dubovik et al., 2014). GRASP also allows to retrieve the volume size distribution from spectral AODs (Torres et al., 2017), assuming a complex refractive index (i.e. 1.50+0.025) and a bimodal lognormal particles size distribution with fixed modal widths."

Discussion on the apparent AOD-H2O relationship (Line 350) should address cloud contamination. Thin clouds can be mistaken as high AOD and are often coincident with high water vapor. How much of the data shown in Figure 10 were taken under clouds? How did you determine that?

**Answer:**

Optically thick clouds are automatically excluded from measurements due to PLASMA2 inability to track the Sun. If optically thin clouds are included in our measurements, the retrieved Angström exponent should clearly demonstrates the presence of cloud droplets above the Sun-photometer with $\alpha_{533-855}$ values below 1.2 (Roelofs and Kamphuis, 2009). However, PLASMA2 $\alpha_{440-870}$ do not show values below 1.8, which is typical of fine mode aerosol as shown by **Figure RC1.2**.

[Figure]

**Figure RC1.2 : Column water vapour in function of AOD and Angström exponent retrieved from PLASMA2 measurements.**

As for the comparison of the calculated DRE values with other estimates, cloud fraction and intentional sampling bias should be mentioned in the place where the comparison is described (Section 4.6), without waiting until Lines 446 and 443. It is worth explaining and emphasizing why these two factors weaken the usefulness of the comparison.

**Answer:**

We mentioned in this section that the OSIRIS DRE estimates were performed for full cloudy scenes (at least at the scale of the OSIRIS image: 20 by 20 km$^2$) and cloud fraction lower than 1 would have led to lower DRE.

We recall that the south-east Atlantic Ocean was overlaid by one of the planet's major stratocumulus cloud deck for the considered time period and that cloud cover with high fractional cloud cover is frequent. 75% of the cloudy scenes sampled with OSIRIS were associated with a geometrical fraction of 1, which is not surprising for the considered area.

The DRE estimates performed with POLDER and SCIAMACHI sensors by De Graaf et al. (2019) were performed for full cloudy scenes for POLDER for COT larger than 3. and for scenes with fractional (effective) cloud cover for SCIAMACHY larger than 0.3 and COT larger than 3.0. So, the DRE comparisons performed between OSIRIS, SCIAMACHI and POLDER DRE are not so inconsistent, in terms of sampled cloud covers.

Indeed, above cloud instantaneous DRE retrievals in a more complex environment (lower fraction of cloud cover and thinner clouds) needs additional information as cloud fraction and cloud microphysics. Comparison between sensors will also be completed by an analyse on the impact of the spatial resolution, which could affect the DRE. These as been re-demonstrated recently on the works of Cochrane et al. (2019) and De Graaf et al. (2019b) respectively, and present interesting analyses for further works.De Graaf et al. (2019b) have merged several satellite measurements with different spatial resolutions. Low resolution smooths signal and ignore under-pixel variation of the cloud albedo, which tends to lead to underestimation of cloud fraction and cloud albedo, whereas high resolution can under-estimate 3D-effects at the edge of the clouds.

We also added a discussion in the manuscript to allow to the reader the possibility to appreciate the differences observed between OSIRIS, POLDER and SCIAMACHI DREs, with regards to the spatial resolution of each sensor.

**Modifications:**

More details have been added to the DRE discussion.

l.44: "The high absorbing load of AAC combined together with high cloud albedo leads to unprecedented DRE estimates, higher than previous satellite-based estimates. The average AAC DRE calculated from the airborne measurements in the visible range is +85 W m$^{-2}$ (standard deviation of 26 W m$^{-2}$) with instantaneous values up to +190 W m$^{-2}$ during intense events. These high DRE values, associated with their uncertainties, have to be considered as new upper cases to evaluate the ability of models to reproduce the radiative impact of the aerosols over the South-East Atlantic region."

l.474: The whole DRE discussion has been clarified: "These mean DRE are higher than previous retrievals in the region. […] The relative low aerosol loading observed on 12 September (AOD of 0.12-0.18 at 865 nm) are still associated with significant values of DRE (+65 ± 25 W.m-2 on average) mainly because of the high COT (20-30) retrieved on 12 September."

We thank the reviewer for the below corrections.

Minor suggestions follow.

Line 3 Remove the period.

L. 16 Remove the first "of".

L. 22 Spell out AAC.

L. 25 Don't spell out AAC.

L. 27 "show" should precede "a".

L. 44 Replace telluric with terrestrial.

L. 95 "coexist" should read "coexists".

L. 105 Sect. 4 should read Sect. 5.

L. 119 Put "m" after 15.

L. 120 Remove the vertical bar.

L. 133 "the" should precede "airborne".

L. 140 "wild" should read "wide".

L. 149 Did you mean to put "a" before "few"?

L. 184 Put "to" after "referred".

L. 190 "overcoast" should read "overcast".

L. 261 The backscattering of 1.10-6 m-1sr-1 is not necessarily low. Shinozuka et al. (2020) used a threshold one fourth as large to identify smoke. How about noting the value under which 95% of the Sept 12 data reside, instead of 100%?

Indeed, it was a mistake to take this value as a reference. Because there is no need for this ABC value here, it has been removed.

L. 346 Drop s from "coasts".

L. 355 End the sentence with a period after "area". Capitaliza b in "biomass".

L. 406 Insert "horizontal" before "resolution".

L. 411 Put an s to "contribute".

---

## Author Comment (AC2) · 26 Feb 2021

First, authors would like to thank the reviewer for his/her interest on this work and his/her constructive comments. We believe they clearly helped highlighting the main goal of the paper and better fit with the ACP scope.

In this study, the authors document imager and sun-photometer retrievals of above-cloud aerosol depth, single-scattering albedo, Angstrom exponent, refractive index, and size distribution made during the AEROCLO-sA aircraft field campaign over the south-eastern Atlantic. Retrievals are compared to similar measurements by similar aircraft instruments and ground-based sun-photometers. Estimates of cloud optical depth and column water vapour are also discussed.

The paper is straightforward, discussing each variable in turn with a few supporting figures, with brief, often speculative, explanations for differences between flights and instruments. The paper would therefore be a useful citation for future users of the data. But the paper reads too much like a data description document and does not align with the ACP scope of publishing "studies with general implications for atmospheric science". However, I think it is possible for the paper to actually tell a story that would make it a valuable contribution to ACP. I also find that several conclusions are insufficiently supported by the analysis. For these reasons, which are detailed below, I recommend major revisions to improve the discussion.

In previous works, the method has been validated from satellite measurements over the SEA region but never used on airborne measurements offering the access to high spatial resolution and validation with combined measurements. Airborne measurements performed during the campaign is thus an opportunity to validate for the first-time aerosol above cloud algorithms applied on airborne polarimetric measurements, but also, as suggested by the reviewer, to focus on the quantification of the direct radiative effect of aerosol above cloud in the SEA region, and the associated uncertainties.

In the following, authors answer to the reviewer and list modifications made to the paper.

**1 Main comments**

• The paper covers a succession of aerosol and cloud variables, without a clear scientific question to answer. That makes it a frustrating read. One solution would be to make the calculation of the direct radiative effect (section 4.6) the objective of the paper. To do that calculation, one needs to characterise aerosols, clouds, etc., which motivates the need for sections 4.1 to 4.5. Note that an analysis of uncertainties in direct radiative effect, propagated from the retrieved optical properties, would be required.

**Answer:** Thanks to this constructive suggestion, the main objective of this work has been clarified in the document. To access direct radiative properties of above cloud aerosols, one need to retrieve scattering and absorbing properties of aerosols as well as cloud properties with high accuracy. The main structure of the manuscript remains unchanged since the complete characterisation of aerosol and cloud properties is necessary to understand DRE values, but clarifications of the main objective are added in every section and more discussion on DRE results is added.

In addition, we also performed an error budget for the DRE calculation and included these new calculations in our study. It accounts for the uncertainties associated with each aerosol and cloud parameters needed to compute the AAC DRE (AOD, particles size, SSA, COT and droplets effective radius). It also accounts for the variability of the water vapor concentration observed during the flights. The sensitivity of the DRE to all of these quantities is also now discussed. For the aerosol parameters, the DRE errors budget is based on the observational uncertainties obtained from the comparisons performed between OSIRIS, PLASMA2 and other available sensors. We think that this approach allowed to implement a realistic and rigorous DRE errors budget. The equations used for the calculations of the DRE were reported in annexes (see Annexes A and B in the new version of the paper), for the sake of clarity and ease of reading, whereas the sensitivity study and the discussion parts were included in the core of the text.

**Modifications:**

We reported here below the modifications that clarify the new main goal of the paper, which is now the calculation of the direct effect of the above cloud aerosols.

Title: The title was changed to better correspond to the main goal of the study: "Aerosol above cloud direct radiative effect and properties in the Namibian region during AEROCLO-sA field campaign: 3MI airborne simulator and sun-photometer measurements.".

Figure 7a has been changed with more realistic uncertainties for each parameter and Annexe A and B have been added to clarify calculations.

DRE uncertainties for each analysed flight have been added to Figure 12 and discussed in the main text.

l.19: "The study aims to retrieve the aerosol above cloud Direct Radiative Effect (DRE) with well-defined uncertainties.".

l.33: "The Single Scattering Albedo (SSA) is one of the most influencing parameters on the AAC DRE calculation that remains largely uncertain in models.".

L43: "The detailed characterization of aerosol and cloud properties, water vapor and their uncertainties obtained from OSIRIS and PLASMA2 measurements allows to study their impacts on the aerosol above cloud DRE.".

l.57: The whole introduction has been clarified and reorganised in order to better correspond to the main goal of the study: "Aerosol particles significantly impact the radiative budget of the Earth. […] Therefore, the new observation capabilities proposed by the airborne instrument OSIRIS give an interesting opportunity to characterise both cloud and absorbing particles in order to retrieve the aerosol DRE with high accuracy. Results are benefit to constrain climate models and satellite retrievals in a climate-sensitive region (Mallet et al., 2019).".

l.118: "In this paper we present aerosol and cloud retrievals performed over the SEA region essential for the calculation of the aerosol DRE.".

l.248: "The primary parameter influencing the aerosol above cloud DRE is the Aerosol Optical Depth (AOD) of the aerosol layer lofted above clouds. Above clouds AODs were measured directly with the sun-photometer PLASMA-2 during specific parts of the AEROCLO-sA flights. These high accurate AOD measurements allows for the validation of OSIRIS above clouds AODs as a first step of the study. AERONET measurements are also used in this section to depict the general AOD variability observed during the field campaign.".

l.313: "Aerosol size can have a significant impact on DRE calculation since it mainly controls the spectral dependency of the aerosol optical thickness. The Angström exponent is a parameter primarily indicative of the particles size. The Angström exponent retrieved with OSIRIS is evaluated against PLASMA measurements and particles size retrievals in this section."

l.363: "The SSA is one of the three most important parameters influencing aerosol DRE calculation with the above cloud AOT and the cloud albedo (Peers et al., 2015). The retrieval of this parameter is still subject to large uncertainties in this region (Peers et al., 2016, Pistone et al., 2019). The retrieval of the SSA from passive remote sensing technics depends on the microphysical assumption. This parameter is primarily driven by the aerosol absorption (i.e. imaginary part of the complex refractive index) and, to a lesser extent by the particles size."

l.404: "The biomass burning aerosol layers transported in the studied region are accompanied by water vapor, with potential significant effects on the radiative budget (Deaconu et al., 2019). It is therefore needed to consider the contribution of water vapour in the studied region to establish an accurate estimate of the aerosol DRE and related uncertainties."

l.298: "The grey zones correspond to the OSIRIS retrieval uncertainty (as described in **Annexes A and B).**"

l.403: More details are brought to the direct radiative effect section: "The Direct Radiative Effect (DRE) calculations are performed over the solar spectrum (0.2-4 microns) with the radiative transfer code GAME (Dubuisson et al., 1996). […] The relative low aerosol loading observed on 12 September (AOD of 0.12-0.18 at 865 nm) are still associated with significant values of DRE ($+65 \pm 25$ W.m$^{-2}$ on average) mainly because of the high COT (20-30 at 550nm) retrieved on 12 September.".

l.417: "A new set of data of cloud and above-cloud aerosol properties allows to retrieve local aerosol above cloud DRE in the South-Eastern Atlantic region, where important bias persist between climate models. The detailed characterisation of the atmospheric content from the polarimetric imager OSIRIS offers the opportunity to study the sensitivity of aerosol and cloud parameters to the local radiative budget."

L.447: The full DRE section has been rewritten according uncertainty study and more discussions: "The DRE calculations are performed […] These high DRE results, associated with their uncertainties, have to be considered as upper cases to evaluate the ability of the models to reproduce the aerosol radiative impact over this region."

l.509: "allows to retrieve local aerosol above cloud DRE in the South-Eastern Atlantic region, where important biases persist between climate models for both the amplitude and the sign of the aerosol radiative perturbation. The detailed characterisation of the atmospheric particles content achieved from the polarimetric imager OSIRIS allowed to study the sensitivity of the local radiative budget to the main optical aerosol and cloud parameters."

Finally, the conclusion has been modified to better correspond with the main goal of the work:

L552: "The mean AEROCLO-sA instantaneous DRE value is +85 W m$^{-2}$ for AAC with mean uncertainties of ± 24 W.m$^{-2}$. We performed a detailed error budget for the DRE. Errors for the aerosol parameters (AOT, Angstrom exponent and SSA) were controlled based on comparisons of data from various sensors. This approach allowed a realistic calculation of the DRE uncertainties. This error budget also accounts for the variability observed for the water vapor during the flights and for the potential changes in the cloud particles microphysics. Our DRE estimates agree with previous studies indicating a strong positive aerosol forcing over the region (De Graaf et al., 2019b, 2020). Obtained DRE are generally higher than previous satellites ones, mainly because of the exceptional atmospheric conditions encountered during the flight (i.e. combination of high absorbing aerosol loads with high cloud albedo). As compared to previous satellite and modeled DRE obtained in the region, the airborne polarimeter used in the present study demonstrates high accuracy on the retrieved above cloud AOD, the absorption property and the cloud optical thickness in the visible-near-infrared domain. These well-defined aerosol and cloud properties have to be considered to evaluate the ability of models and satellites to reproduce locally high instantaneous DRE."

L565: "In conclusion, the airborne multi-viewing, multi-channel, multi-polarisation measurements in the region allow us to obtain aerosol and cloud properties with well characterized uncertainties as well as their sensitivity to aerosol above cloud DRE. Such findings are valuable to constrain climate models and also evaluate satellite retrievals as future 3MI measurements (Marbach et al., 2015). The high spatial resolutions, offered by the airborne polarimeter OSIRIS will allow to accurately estimate the DRE and the cloud properties and variability within regional model grids. Spectral extension of the OSIRIS algorithm will incorporate additional UV data from the airborne micro-polarimeter Ultra-Violet (MICROPOL), also operated on the Safire Falcon-20 during AEROCLO-sA, as well as additional spectral bands in the middle-infrared (up to 2.2 microns). These will benefit to the characterization of the spectral absorption of the aerosol, linked to their chemical composition, and for the retrieval of the cloud microphysics, which is crucial for the study of the aerosol and cloud interactions. Lidar LNG profiles combined to OSIRIS data will allow to further evaluate the heating rate profiles of aerosol above clouds and to study the cases of interaction when aerosol and clouds are in contacts at the cloud top. Further investigation based on the combination of this new set of observations and regional models, as described in Formenti et al. (2019), will be of greatly interest for such studies leaded in the SEA region."

• The paper opens with a tantalising description of a new instrument, 3MI (lines 83- 92), and concludes that 3MI improves the "definition" (unclear what is meant by that word) of above-cloud aerosol properties (lines 449-451). But the discussion does not clearly identify what new abilities 3MI brings compared to its predecessor, POLDER, and how the AEROCLO-sA field campaign helps demonstrate these new capabilities. The discussion needs to support that conclusion. What did the authors do that could not be done before? What has been done better?

**Answer:** The proposed work is based on the airborne prototype OSIRIS of the future satellite instrument 3MI. As a successor of the POLDER instrument, 3MI is a polarimetric imager that will be in a polar orbit. In addition, the 3MI measurements will be realised both in the visible and in the middle infrared spectral domain (up to 2.2 microns) with higher spatial resolution and a larger field of view than proposed by POLDER.

We removed the term "definition" which was effectively unclear.

The present paper now focuses on the retrieval of the main optical parameters required to compute the DRE of biomass burning aerosol above clouds over the solar spectrum. These parameters are: the spectral AOD, particle size, SSA, COT, and droplets effective radius. As a secondary goal, this study also allows to evaluate (for the first time) the capabilities of the POLDER algorithm developed for AAC scenes to quantify the aerosol properties above clouds.

A second paper will be focused on the use of the shortwave infrared channels of OSIRIS offered by the new capabilities of the 3MI instrument. This second paper will indeed allow to evaluate the new capabilities of 3MI for climate studies. Mainly, the inclusion of additional measurements in the middle-infrared will help to better characterize the cloud droplets microphysics. In the context of AEROCLO-sA, this is primarily interesting for the evaluation of the potential effects of aerosol on the cloud microphysics. Secondly, we plan to adapt the GRASP algorithm (Dubovik et al., 2012) to the OSIRIS visible and middle-infrared measurements. This algorithm is the official operational algorithm selected by ESA for the retrieval of aerosol properties with 3MI in clear-sky conditions. This will allow to evaluate the benefit of these new spectral bands for the retrievals of the coarse mode particles and surface properties.

In the conclusion section of the new version of the paper, we now clearly indicate that the new peculiar spectral abilities of 3MI will be evaluated with OSIRIS in future paper. We made clear that this is a perspective.

**Modifications:**

l.26: "The combined airborne measurements allow for the first time the validation of Aerosol Above Cloud (AAC) algorithms previously developed for satellite measurements".

l.86: "In order to retrieve aerosol DRE above cloud in the SEA region with well-defined uncertainties, which is needed to evaluate climate models, one need to characterize aerosols and cloud optical and microphysical properties."

l.565: "In conclusion, the airborne multi-viewing, multi-channel, multi-polarisation measurements in the region allow us to obtain aerosol and cloud properties with well characterized uncertainties as well as their sensitivity to aerosol above cloud DRE. Such findings are valuable to constrain climate models and also evaluate satellite retrievals as future 3MI measurements (Marbach et al., 2015). The high spatial resolutions, offered by the airborne polarimeter OSIRIS will allow to accurately estimate the DRE and the cloud properties and variability within regional model grids.".

• The abstract reads that "Combination between water vapour and the strong positive aerosol forcing over the region explains possible feedbacks on cloud development." (lines 40-41) What does that mean? Which part of the discussion supports that statement?

**Answer:** The relation between cloud development and the radiative impacts of the smoke layer transported above clouds, with the "smoke layer" meaning a combination of water and BBA aerosols, is an open discussion and link to findings from the work of Deaconu et al. (2019) in this region. Our study does not focus on this specific problem. However, our finding does not contradict the observations made in Deaconu et al., (2019) since we found (1) a strong and positive direct radiative effect for the above cloud aerosols and (2) that we measured significant amount of water vapor collocated with the BBA.

This sentence was removed from the abstract. We added a sentence in the conclusion section to underline that our observations confirm that AAC BBA are indeed collocated with water vapor, which does not contradict the latter previous study.

**Modifications:**

l.540: "As already noted by previous studies (Deaconu et al., 2019, Pistone et al., 2019), water vapour concentration and aerosol loading estimated above clouds are generally correlated in this region. These observations as confirmed by the sun-photometer measurements performed by the airborne PLASMA. So, our observations do not contradict previous studies indicating that both BBA aerosols and water vapor have to be considered together to investigate the total radiative impacts of smoke plume.".

• The authors find a consistent 10% disagreement between aerosol optical depth from OSIRIS and PLASMA (lines 246, 259, 411). But the implications of that disagreement are never discussed. Is that something to worry about? Is that a systematic bias? What causes it?

**Answer:**

This bias mainly observed at 865 nm can be attributed to the presence of aerosol coarse mode particles whereas our retrievals only consider fine mode particles. Indeed, on 12 September measurements, PLASMA2 inversion of the particle size distribution shows higher coarse mode particle concentrations (Figure RC2.1) than other flight

measurements. This is probably due to some increase in the wind speed at the surface that uplifted some dust for this day and also because the flight was performed straight along the Namibian coast. Thanks to PLASMA retrievals, we estimated the AAC coarse mode AOD to be equal to 0.04 at 670 nm during the loop performed above the clouds on 12 September. For the flights performed on the 7 and 8 of September, the departures observed between OSIRIS and PLASMA for the AOT at 670 nm are of about 0.01 (see table 1), which is the sun-photometer measurements accuracy. So, there is no systematic bias in the OSIRIS AOT retrievals. Based on the PLASMA and AERONET retrievals, we can affirm that the AAC coarse mode AOT can be safely neglected in our DRE calculations for all the flights of AEROCLO-sA, excepted on 12 September. The coarse mode AOD measured on 12 September limits the relative accuracy of our OSIRIS AOT retrievals at 670 nm to 10 %. For the calculation of the DRE uncertainties, we increased our AOT retrieval error to account for this observation (see Annexe B).

[Figure]

**Figure RC2.1 : Retrieved volume particle size distributions at different altitudes from PLASMA2 measurements and from the AERONET station at Windpoort, Namibia, for the descent in loop of 12 September 2017.**

**Modifications:**

We included the following discussion in the new version of the manuscript for sake of clarification:

l.281: "OSIRIS AOD is slightly lower than the PLASMA one on 12 September by 0.04 for the AOT at 670 nm. This low bias can mainly be attributed to the neglect of coarse mode particles in the OSIRIS algorithm, which only considers fine mode particle to model the radiative properties of the aerosol biomass burning aerosols lofted above clouds. Indeed, on 12 September measurements, PLASMA2 inversion of the particle size distribution shows higher coarse mode particle concentrations (not shown) than for other flight measurements. This is probably due to some increase in the wind speed at the surface that uplifted some dust for this day and also because the concerned flight was performed straight along the Namibian coast. Thanks to PLASMA retrievals, we estimated the AAC coarse mode AOD to be equal to 0.04 at 670 nm during the loop performed above the clouds on 12 September. For the flights performed on the 7 and 8 of September, the departures observed between OSIRIS and PLASMA for the AOT at 670 nm are of about 0.01 (see table 1), which is the sun-photometer measurements accuracy. So, there is no systematic bias in the OSIRIS AOT retrievals. Based on the PLASMA and AERONET retrievals, we can affirm that the AAC coarse mode AOT can be safely neglected for the DRE calculations for all the flights of AEROCLO-sA, excepted on 12 September. The coarse mode AOD measured on the 12 of September limits the relative accuracy of our OSIRIS AOT retrievals at 670 nm to 10 %. For the calculation of the DRE uncertainties, we increased our AOT retrieval error to account for this observation (see Annexe B)."

l.302: "Again, the low bias in the OSIRIS AOD retrievals (at 870 nm) is likely due to the presence of few coarse mode particles, as previously discussed."

**2 Other comments**
• Lines 49-50: Need a reference for that statement. Note that the conclusions of Bond et al. (2013

doi:10.1002/jgrd.50171) have been challenged, see the discussion in section 5 of Bellouin et al. (2020 doi: 10.1029/2019RG000660)

**Answer:** New analyses developed by Bellouin et al. (2020) reveal Black Carbon contributions overestimated on absorbing AOD in remote areas by previous studies. Their contribution to the global climate is thus still a challenge since highly positive forcing is observed above cloud in the region through different point of view.

**Modification:**

The paragraph on Black Carbon in the introduction section has been modified to lead the discussion towards the aerosol above cloud DRE. Nevertheless, it is important to site Bellouin et al. (2020) work to keep in mind the debate on the impact of aerosol absorption on the South-Eastern Atlantic climate.

l.65: "The SEA region presents therefore a unique opportunity to study aerosol-cloud-radiation interactions and the impact of highly absorbing particles from biomass burning in central Africa, which are still debated (Bellouin et al., 2020). AeroCom study (Zuidema et al., 2016) demonstrates a net aerosol DRE from cooling to warming in this region. Indeed, Aerosol Above Cloud (AAC) highly contributes to climate uncertainties and very few methods currently allow the retrieval of a detailed model of their properties (Waquet et al., 2013b, Knobelspiesse et al., 2013, Peers et al., 2015)."

• Line 117: I would not call a 35% reduction in data amount slight!

**Answer:** Indeed, 35% represents the selected cloudy cases for the retrievals. We recalled that OSIRIS retrievals can only be performed at high altitude (e.g. > 7 km). Authors believe that indicating the percentage of optimum cloudy cases will be more relevant here.

**Modification:**

l.139: "The selected cases for OSIRIS inversions for aerosol above cloud represent 76% of cloudy measurements at high altitude (> 7 km)."

• Lines 177-181 and Figure 3: I understand that Figure 3 is there to illustrate the retrieval process, but it would be useful to have an idea of the outcome. Which "model" fits the data best? With what optical parameters?

**Modification:** Aerosol Optical Depth, Angström exponent of the aerosol model and Single Scattering Albedo were added to the legend of Figure 3.

Legend Figure 3: "The main aerosol properties retrieved for this case are: AOD = 0.74 at 670 nm, α490-870 = 1.82, and SSA = 0.87 at 670 nm."

• Lines 184 and 191: Does PLASMA2 onboard the aircraft performs the almucantar scans used by the AERONET inversion algorithm? Or are the size distributions derived in other ways?

**Answer:** This second version of the PLASMA instrument does not perform almucantar scans as AERONET Sun-photometers. At this occasion, the aerosol size distributions are retrieved from GRASP-AOD algorithms based on spectral AOD analyses (Dubovik et al., 2014, Torres et al., 2017), as well as the AOD contribution of each aerosol mode. As described from line 192, this needs assumptions on the complex refractive index and a bimodal lognormal volume size distribution.

**Modification:**

l.213: "Because PLASMA2 does not allow to perform almucantar measurements as AERONET Sun-photometers, the fractions of fine and coarse mode AOD was derived using the PLASMA2 spectral AOD measurements using the Generalized Retrieval of Atmosphere and Surface Properties (GRASP) algorithm (Dubovik et al., 2014). GRASP also allows to retrieve the volume size distribution from spectral AODs (Torres et al., 2017), assuming a complex refractive index (i.e. 1.50+0.025) and a bimodal lognormal particles size distribution with fixed modal widths.".

• Lines 189-191: This statement is confusing. What do the authors mean by "low level flight" in this context? Near the ocean surface or near the cloud top?

**Answer:** The goal of the PLASMA2 phase measurements was to include the maximum of the atmospheric column on the measurements. Hence, in clear sky conditions, "low altitude flights" means the closest to the surface (around 1 km height above ground level. In cloudy conditions, "low altitude flights" means at cloud top which allows comparison with aerosol above cloud observations from OSIRIS. Indeed, the term was not detailed enough in the text and clarifications were added.

**Modification:**

l.208: "During AEROCLO-sA, several low-level flights were performed, typically near the cloud top when measurements were performed over the ocean and, near the ground, under clear sky conditions, when the low-level flights were performed over desert sites. PLASMA2 measurements performed at high altitude allowed to characterize the residual columnar AOD above the aircraft. This quantity was subtracted to PLASMA2 AODs measurements performed at low altitude for the sake of comparison with OSIRIS retrievals. Because PLASMA2 does not allow to perform almucantar measurements as AERONET Sun-photometers, the fractions of fine and coarse mode AOD was derived using the PLASMA2 spectral AOD measurements using the Generalized Retrieval of Atmosphere and Surface Properties (GRASP) algorithm (Dubovik et al., 2014). GRASP also allows to retrieve the volume size distribution from spectral AODs (Torres et al., 2017), assuming a complex refractive index (i.e. 1.50+0.025) and a bimodal lognormal particles size distribution with fixed modal widths.".

• Line 252: "robust independently on the aerosol loading." Too strong a statement considering the limited range of aerosol optical depth shown in Table 1.

**Answer:** The reviewer is right about the limited range of AOD recorded during the AEROCLO-sA campaign. AOD retrieved are from moderate to high with values between 0.36 and 0.74 at 670 nm. The statement has been adapted to this comment.

**Modification:**

l.278: "[…] good agreement between OSIRIS and PLASMA2 for moderate to high aerosol loading (AOD from 0.36 and 0.74 at 670 nm)**"**

• Line 355: What do the authors mean by "wood moisture"? The moisture emitted by evapotranspiration of forests?

**Answer:** Biomass burning usually generates water vapour in the atmosphere because of the process of combustion that generates water vapor. The type and efficiency of the combustion process then depends on the humidity present and contained in the vegetation (Betts and Silva Dias, 2010; Sena et al., 2013).

We clarified this point.

**Modification:**

l.413: "This correlation might be also the result of the direct emission of water vapour due to the fires themselves (Betts and Silva Dias, 2010; Sena et al., 2013). The water vapor amount is quite variable for one flight to another varying between 0.7 and 2.7 g.cm$^{-2}$. We estimated the mean water vapour amount to be equal to 1.7 g.cm$^{-2}$ for the AAC scenes sampled during AEROCLO-sA. Note that dropsonde measurements were used to supplement the PLASMA2 data in order to estimate the amount of water vapor within the cloud layer. Finally, one can note that there is no correlation for the second flight of 12 September between AODs and water vapor measurements. Low water vapour amount (below 1 g cm$^{-2}$) and high AOD values (>0.7) were observed together for this flight. We do not have full explanation for this contradictory observation. These observations were obtained for an in-land location (Etosha Pan) and we assume that this area could be associated with dryer air masses than the ones sampled over the oceanic regions.".

• Line 359: "This particularity" – what does it refer to? The absence of correlation, or the low water content? It is unclear whether the different behaviour seen in Figure 10 for flight 1209 F2 is in fact understood.

**Answer:** "This particularity" refers to the absence of correlation compared to other flights. This sentence has been clarified.

**Modification:**

l.418: "Finally, one can note that there is no correlation for the second flight of 12 September between AODs and water vapor measurements. Low water vapour amount (below 1 g cm$^{-2}$) and high AOD values (>0.7) were observed together for this flight. We do not have full explanation for this contradictory observation. These observations were obtained for an in-land location (Etosha Pan) and we assume that this area could be associated with dryer air masses than the ones sampled over the oceanic regions.".

• Line 383: "DRE calculations are performed online". What does that mean?

**Answer:** "Online" means without any previously calculated look up tables as used in previous works on POLDER algorithms (Peers et al., 2015). This new approach allows sensitivity study on the calculation of DRE according aerosol and cloud properties.

• Line 408: "extreme environment". In what sense? Having a large aerosol loading probably makes retrieving their optical properties easier.

**Answer:** Here, « extreme » means the highest part of the AOD range observed in this region, not a term of difficulty. It was not clear. The sentence has been changed.

**Modification:**

l.518: "Measurements were performed in the highest range of retrieved AOD in this region above a semi-permanent stratocumulus cloud deck."

• Line 419: "significantly impacts the climate". This conclusion is outside the scope of the study, so should either be supported by references, or made conditional.

**Answer:** We agree. We focus on the estimate of the direct radiative effect that are significant in our case studies. The term climate is indeed appropriate for the estimate of aerosol direct forcing and long-term analysis.

**Modification:**

l.529: "Biomass burning aerosols transported over the South-eastern Atlantic Ocean represent a high absorbing effect which significantly impacts the direct radiative effect."

**3 Technical comments**

• Line 17: analyse of -> analyse

**Modifications:** Done

• Abstract: Some acronyms are defined, but others (OSIRIS, PLASMA2, POLDER) are not. Need to make it consistent.

**Modifications:** Definitions of the acronyms have been harmonized.

• Line 27: a show -> show a

**Modifications:** Done

• Line 44: "telluric": English speakers tend to prefer "terrestrial"

**Modifications:** Done

• Line 80: change on -> change in

**Modifications:** Done

• Figure 1: the coastline could easily be mistaken for a flight track! Perhaps set a Google Earth background, or blue over ocean, or something like that?

**Modifications:** Thanks. The Figure 1 was modified.

• Line 133: airborne the -> airborne

**Modifications:** Done

• Line 190: overcoast -> overcast

**Modifications:** The sentence has been changed.

• Line 209: form -> form

**Modifications:** Done

• Line 256: north cape -> northbound heading

**Modifications:** Done

• Caption of Figure 7: need to state that panel b is from OSIRIS measurements.

**Modifications:** This has been specified in the legend.

• Line 315: spatial -> spatially

**Modifications:** Done

---

## Author Response (AR2)

Authors thank the reviewer for this interesting discussion on water vapour impacts on aerosol observations which is still a well-known concern. In the following, authors answer reviewers' comments and specify modifications realised in the manuscript.

I find the discussion on water vapour in Section 4.4 ambiguous. Do the authors vary water vapour amounts in their ADRE calculations? Section 4.6 suggest that they don't, and indeed I do not think that they should -- the water would probably be in the atmosphere anyway, as it is associated with the air mass, not the aerosols. (I doubt that fires would produce enough water to change atmospheric water vapour amounts significantly.) And what would be the reference ("no aerosols") value anyway?

I agree that the presence of water influences ADRE via hygroscopic growth -- but that is implicitly accounted for by the retrieved AOT and SSA.

**Answer:** Discussion on water vapour on section 4.4 allows to remind lecturers the importance of water vapour on radiative budget analyses. Whereas water vapour concentrations between 0.7 and 2.7 g cm$^{-2}$ are observed locally from dropsondes during the campaign, this effect is secondary in the visible range. At this occasion, a constant value of 1.7 g m$^{-2}$ is used, based on mean profiles obtained from dropsondes during the campaign. More realistic calculations of the ADRE are realised using a more realistic reference atmosphere. To consider water vapour variations during the campaign, a standard deviation of 1 g m$^{-2}$ is included in the ADRE uncertainty calculation. This was mentioned in section 4.6 and in the Annexe B.

l.461: "The total amount of columnar water vapor was fixed to a value of 1.7 g.cm-2. We assume an error of ± 1 g.cm-2 in accordance with the PLASMA observations for this quantity."

l.880: "Additional errors terms were also added to account for the variability in the water vapor amount (standard deviation of 1 g cm-2) and for the cloud droplet effective radius (standard deviation of 2 microns)."

I am also unconvinced by the statement "These observations were obtained for an in-land location (Etosha Pan) and we assume that this area could be associated with dryer air masses than the ones sampled over the oceanic regions." -- The water vapour in the transported aerosol plumes comes from the land, so assuming land=dry ocean=moist does not seem correct in this case.

**Answer:** The origin of water vapour retrieved in our measurements is also discussed but needs more analyses on transportation and local emissions to be refined. This sentence brings hypothesis which can explain measurements without any certainty. A recent work brings new analyses on the water vapour concentration link to the biomass burning plume in this region (Pistone et al., 2021). Based on aircraft measurements and model simulations, authors demonstrate that the water vapour concentration which is linearly correlated with CO concentration, may not originate to BB emissions. This new result is added to the discussion.

**Modification:**

l.421: "Based on aircraft measurements and model simulations in the South-Eastern Atlantic region, a recent study demonstrates that the water vapour concentration which is linearly correlated with CO concentration, may not originate to BB emissions (Pistone et al., 2021). Hence, the meteorology seems to mainly drive the amount of water vapour in the atmosphere in this region."

Finally, Figure 12 needs a legend to explain the colours. Are they the same as in Figure 11b?

**Answer:** Figure 12 is based on the same representation than Figure 11b.

**Modification:**

The Figure 11b legend was added to Figure 12.